# COMPETITION IS THE KEY: A GAME THEORETIC CAUSAL DISCOVERY APPROACH

## ABSTRACT

We introduce a **game-theoretic reinforcement learning framework** for causal discovery in which a DDQN agent *competes* with a strong baseline (GES or GraN-DAG) and always *warm-starts* from the opponent's graph. This yields three key guarantees: the learned graph is *never worse* than the warm start, warm-starting *accelerates convergence*, and with high probability the method selects the best candidate when $n$ is large enough. Formally, if $n \geq \frac{8L^2}{\Delta_n^2} \log\left(\frac{2|C|}{\delta}\right)$, then with probability $1 - \delta$ the algorithm recovers the population-optimal graph. Here $L$ is a Lipschitz constant of the score function, $\Delta_n$ is the empirical gap between the best and second-best candidate scores, $|C|$ is the number of candidate graphs considered, and $\forall \delta \in (0, 1)$ is the failure probability. To our knowledge, this is the first finite-sample consistency result for an RL-based causal discovery method. Empirically, DDQN-CD matches or outperforms GES and GraN-DAG on standard benchmarks (Sachs, Asia, Alarm, Child, Hepar2) and scales to large graphs (Dream $\sim$100, Andes $\sim$220 nodes). Our results demonstrate that RL-based discovery can be simultaneously *provably safe*, *sample-efficient*, and *scalable*, helping bridge the gap between theoretical guarantees and practical performance.

## 1 INTRODUCTION

Randomized controlled trials (Hariton & Locascio, 2018) are widely regarded as the gold standard for causal inference, but in many domains they are infeasible, prohibitively expensive, or ethically questionable (Chen et al., 2023). This limitation has driven sustained interest in causal discovery from observational data, yet every major family of algorithms comes with sharp drawbacks. Constraint-based methods such as PC (Spirtes et al., 2001) and FCI (Spirtes et al., 2001) rely on conditional independence tests, but suffer from instability: a single skeleton error can cascade into widespread orientation mistakes. Score-based methods like GES (Chickering, 2002) optimize likelihood criteria with complexity penalties, but the search is NP-hard, requiring greedy heuristics that can stall under finite samples or model misspecification. Functional causal models (e.g., LiNGAM (Shimizu, 2014), ANM (Hoyer et al., 2008)) guarantee identifiability only under restrictive assumptions, and fail when real data violate them. Continuous optimization relaxations such as NOTEARS (Zheng et al., 2018), DAG-GNN (Yu et al., 2019), and GraN-DAG (Lachapelle et al., 2019) enforce acyclicity through smooth constraints, but are tied to specific surrogate losses, limiting their ability to incorporate arbitrary scores or robustness objectives.

Reinforcement learning (RL) has been proposed as a flexible paradigm for causal discovery. RL-BIC (Zhu et al., 2020) showed that policy-based exploration can outperform GES on several benchmarks, and CORL (Wang et al., 2021) framed node ordering as a Markov decision process. More recently, KCRL (Hasan & Gani, 2022) argued that prior knowledge can be injected via reward-penalty constraints to shrink the search space and accelerate convergence. Yet these methods remain essentially heuristic: RL-BIC exhibits unstable precision-recall trade-offs, CORL generalizes poorly beyond the Sachs dataset, and KCRL leaves open the fundamental question of whether reinforcement learning for causal discovery can be placed on firm theoretical ground.

**This work takes a step in addressing the above limitations.** We propose a Double Q-learning (Van Hasselt et al., 2016) framework for causal discovery, DDQN-CD, that transforms RL-based search into a principled, theoretically controlled procedure. Our framework integrates robust BIC scores (Copula-BIC), warm-starts the search from strong classical opponents such as

GES or GraN-DAG, and enforces feasibility through action masking and edge budgets. Crucially, the algorithm maintains a *champion-challenger* setup: it never returns worse than its opponent, it provably reduces the expected time to reach a local optimum when warm-started, and it offers finite-sample guarantees that the probability of selecting a suboptimal graph decays exponentially with sample size. In short, we move RL-based causal discovery from heuristic exploration to a theoretically grounded optimization framework.

We validate our approach in two regimes. On synthetic data, we directly stress-test the theorem, demonstrating that the probability of mis-selection shrinks with $n$ while the population gap grows. On real benchmarks -Sachs (Zhang et al., 2021), Asia (Lauritzen & Spiegelhalter, 1988), Alarm (Beinlich et al., 1989), Child (Spiegelhalter et al., 1993), Hepar2 (Onisko, 2003), DREAM (Kalainathan et al., 2020), and Andes (Conati et al., 1997) we demonstrate scalability. In particular, Hepar2, DREAM and Andes contain 70, 100, and 220 nodes respectively, where several competing RL-based or continuous methods fail outright, yet DDQN-CD consistently delivers competitive or superior structure recovery. These results establish DDQN-CD as a scalable, theoretically grounded alternative to existing causal discovery algorithms. Our contributions are summarised as follows( ref. Figure 1)

**Organization.** The remainder of this paper is organized as follows. Section 2 surveys prior work on causal discovery, with particular attention to reinforcement learning approaches. Section 3 introduces our DDQN-CD framework, outlining the game-theoretic formulation, reward design, and learning algorithm. Theoretical guarantees are presented in detail in Section 4. Section 5 provides both theoretical verification of Theorem 3 and empirical evaluations on benchmark datasets spanning small, mid-scale, and large networks, accompanied by insights and broader implications. Finally, Section 6 concludes the paper and highlights future research directions. [1]

| Unified RL-based Framework | Theoretical Guarantees | Scalability Across Real Datasets |
|---|---|---|
| • Double Q-learning for causal discovery
• Warm-start with GES / GraN-DAG
• Robust BIC (Copula-BIC) | • Never worse than the warm-start opponent
• Faster hitting time to a local optimum
• Suboptimal selection prob. decays exponentially in $n$ | • **Small** (Asia, Sachs, Lucas): near-perfect (TPR= 1.0, FDR= 0.0 on Asia/Lucas)
• **Mid** (Alarm, Hepar2): balanced; stronger than RL-BIC2, competitive with GES
• **Large** (DREAM, Andes): SHD ↓ 30-40% vs. Gran-DAG at 100-200+ nodes |

Figure 1: Summary of our contributions. DDQN-CD integrates a unified RL-based framework with theoretical guarantees and demonstrates scalability across diverse benchmarks.

## 2 RELATED WORK

Causal discovery has been extensively studied across multiple paradigms, including constraint-based approaches (e.g., PC (Spirtes et al., 2001)), score-based search (e.g., GES (Chickering, 2002)), functional causal models such as LiNGAM (Shimizu et al., 2006; 2011), and continuous optimization frameworks like NOTEARS (Zheng et al., 2018), GOLEM (Ng et al., 2020), and GraN-DAG (Lachapelle et al., 2019). While these methods provide strong theoretical guarantees or computational elegance, they often struggle with scalability, robustness to noise, or the ability to balance precision and recall in large networks.

Reinforcement learning (RL) has emerged as a flexible alternative, framing causal discovery as a sequential decision-making problem. RL-BIC2 (Zhu et al., 2020) introduced actor-critic based search guided by BIC rewards, but it suffers from instability and limited recall. CORL (Wang et al., 2021) cast node ordering as an MDP but is restricted in applicability beyond small datasets such as Sachs. More recently, KCRL (Hasan & Gani, 2022) incorporated domain knowledge via reward-penalty shaping, improving convergence but leaving open questions of scalability.CORE (Sauter et al., 2024) is a scalable RL approach for causal discovery, but its emphasis is on efficient exploration heuristics rather than theoretical guarantees. These works highlight both the promise and the limitations of RL-based discovery. Beyond RL-based methods, recent advances such as GFlowNets (Manta et al., 2023) and neural causal models (Lippe et al., 2021) explore alternative sampling and continuous-optimization paradigms, but do not provide discrete acyclicity control or

---

[1]Our code is available here - Code link

finite-sample guarantees of the type we establish. In contrast, our method leverages Double DQN with opponent warm starts (GES or GraN-DAG) and BIC-based rewards, transforming RL from a heuristic into a scalable, theoretically grounded framework that consistently outperforms across small, mid, and large networks.

# 3 BACKGROUND AND PRELIMINARIES

**DAGs and Causal Discovery.** A directed acyclic graph (DAG) $G = (V, E)$ represents a causal model over variables $X_1, \ldots, X_p$, where $(i \to j) \in E$ encodes a direct causal influence. Two DAGs are *Markov equivalent* if they share the same skeleton and $v$-structures; the corresponding Markov Equivalence Class (MEC) is

$$\text{MEC}(G) = \{G' : G' \text{ has same skeleton and v-structures as } G\}.$$

Under purely observational linear-Gaussian assumptions, $G$ is identifiable only up to its MEC.

**Reinforcement Learning (RL).** We model the search over DAGs as a Markov Decision Process (MDP):

$$\mathcal{M} = (\mathcal{S}, \mathcal{A}, T, R, \gamma),$$

where states $\mathcal{S}$ are DAGs, actions $\mathcal{A}$ are edge add/remove/reverse operations, $T$ is the deterministic transition operator, and $R$ is the BIC score difference $R(G, a) = S_n(T(G, a)) - S_n(G)$. The agent learns an action-value function

$$Q^\pi(G, a) = \mathbb{E}_\pi \Big[ \sum_{t \geq 0} \gamma^t R(G_t, a_t) \Big| G_0 = G, a_0 = a \Big],$$

and follows an $\varepsilon$-greedy policy. This formulation enables long-horizon planning and persistent exploration.

**Game-Theoretic View (Champion–Challenger).** Our procedure admits a game-theoretic interpretation: the warm-start graph $G_{\text{init}}$ plays the role of a fixed *opponent*, while the RL agent proposes *challenger* graphs $\{G_t\}$ by exploring the MDP. The final "winner" is selected as

$$G_{\text{out}} = \arg\max\{S_n(G_{\text{init}}), S_n(G_1), S_n(G_2), \ldots\},$$

and Theorem 3 guarantees that, with high probability, $G_{\text{out}}$ coincides with the best member of the entire candidate set.

We cast causal discovery as a *sequential game* between a reinforcement learning agent and an opponent prior (GES or GraN-DAG), which provides a warm-start graph $A_0$. The agent refines $A_0$ through local edge edits (ADD, REMOVE, REVERSE), restricted to acyclicity and edge-budget constraints. Each move receives a payoff

$$r(A \to A') = \frac{S(A') - S(A)}{p} - \lambda \|A'\|_0 - c,$$

capturing normalized BIC improvement, sparsity, and step cost. Training proceeds via Double DQN with replay buffer and Polyak updates, where the agent selects actions $\varepsilon$-greedily until a stopping criterion is met (Algo. 1). This *champion–challenger* setup guarantees the discovered DAG $\widehat{G}$ is never worse than its opponent, turning strong priors into stepping stones for scalable, accurate causal discovery. For more details, please refer Appendix A

# 4 THEORETICAL GUARANTEES

## PRELIMINARIES

Let $p$ be the number of variables. The finite set of actions $\mathcal{A} \subset \{0, 1\}^{p \times p}$ consists of binary adjacency matrices respecting acyclicity and a configured edge budget $B$. An action is one of $\{\text{add}(i \to j), \text{remove}(i \to j), \text{reverse}(i \to j)\}$ for $i \neq j$. Each episode resets to a warm start graph $\tilde{G}$ and consists of at most $L$ edits.

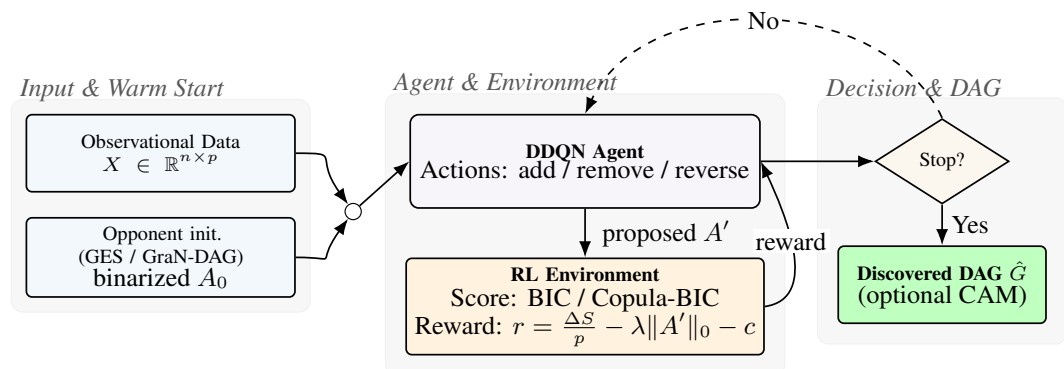

Figure 2: DDQN–CD framework. Observational data and a warm start are merged and fed to a DDQN agent that proposes edge edits; an environment evaluates candidates via BIC/Copula-BIC and returns a reward until the stopping condition triggers, yielding the discovered DAG.

---

**Algorithm 1** DDQN–CD: Double Deep Q–Learning for Causal Discovery with BIC Reward and Opponent Warm Start

---

**Require:** Data $X \in \mathbb{R}^{n \times p}$; opponent flag $o \in \{\text{GraN-DAG, GES}\}$; hyperparameters $(\gamma, \tau, \lambda, c, B, T, E, P)$
**Ensure:** Discovered DAG $\widehat{G}$
1: **Scorer** $S$: use *DiscreteBIC* if $X$ is binary, else *CopulaBIC* (Gaussian BIC after rank–Gaussian transform)
2: **Warm start** $A_0 \leftarrow \text{binarize}(\text{Opponent}(X, o))$     ▷ GraN-DAG or GES output
3: **Actions** on ordered pairs $(i, j)$, $i \neq j$: ADD$(i \to j)$, REMOVE$(i \to j)$, REVERSE$(i \to j)$
4: Mask invalid actions to keep acyclicity and edge budget $\|A\|_0 \leq B$
5: **Reward** for $A \to A'$:

$$r(A \to A') = \frac{S(A') - S(A)}{p} - \lambda \|A'\|_0 - c$$

6: Initialize online $Q_\theta$ and target $\bar{Q}_{\bar{\theta}} \leftarrow Q_\theta$; replay buffer $\mathcal{M}$; best graph $\widehat{G} \leftarrow A_0$
7: **for** $e = 1$ **to** $E$ **do**              ▷ episodes
8:   $A \leftarrow A_0$                ▷ reset
9:   **for** $t = 1$ **to** $T$ **do**            ▷ steps
10:    $m \leftarrow$ valid action mask from $A$
11:    **(action)** with prob. $\varepsilon$: sample $a$ uniformly from $\{k : m_k = 1\}$; else $a \leftarrow \arg\max_{k:\, m_k=1} Q_\theta(A, k)$
12:    **(transition)** If $a$ valid & keeps DAG/budget produce $A'$; otherwise set $A' \leftarrow A$, reward $r \leftarrow -\delta$
13:    **(reward)** If valid, set $r \leftarrow \frac{S(A') - S(A)}{p} - \lambda \|A'\|_0 - c$
14:    Store $(A, a, r, A')$ in $\mathcal{M}$; set $A \leftarrow A'$
15:    **if** $|\mathcal{M}|$ sufficient **then**          ▷ Double DQN update
16:     Sample mini-batch $\{(A^i, a^i, r^i, A'^i)\}_{i=1}^b$ from $\mathcal{M}$
17:     $y^i \leftarrow r^i + \gamma \bar{Q}_{\bar{\theta}}(A'^i, \arg\max_k Q_\theta(A'^i, k))$
18:     Update $\theta$ by one SGD step on $\frac{1}{b} \sum_i \left( Q_\theta(A^i, a^i) - y^i \right)^2$
19:     Polyak target: $\bar{\theta} \leftarrow (1 - \tau)\bar{\theta} + \tau\,\theta$
20:   **if** $e \bmod P = 0$ **and** $S(A) > S(\widehat{G})$ **then**   $\widehat{G} \leftarrow A$
21: **return** $\widehat{G}$           ▷ (Optionally apply CAM pruning post hoc)

---

Given the full dataset $X_{1:n}$, the empirical score is the Bayesian Information Criterion (BIC):

$$S_n(A) = \sum_{t=1}^n \ell(A; X_t) - \frac{1}{2} k(A) \log n,$$

where $\ell$ is the log-likelihood per sample and $k(A)$ is the parameter count. The algorithm maintains a *champion* snapshot $\widehat{G}$ with the largest score seen and ultimately returns the better of $\widehat{G}$ and the opponent $\tilde{G}$. All the necessary assumptions are mentioned in Table 1. All our guarantees (Theorems 1–3) are MEC-level

Table 1: Summary of Theoretical Assumptions

| Assumption | Formal Description |
|---|---|
| **A1: Finite Feasibility** | Acyclicity and the edge budget $B$ are enforced by masking, so the feasible set $\mathcal{A}$ is finite. Each episode lasts at most $L < \infty$ steps. |
| **A2: Persistent Exploration** | The policy explores with probability at least $\varepsilon_\star > 0$. When exploring, the action is chosen uniformly from all valid actions. |
| **A3: Warm Start** | Every episode initializes at the opponent DAG $A_0 = \tilde{G}$. |
| **A4: Champion-Challenger** | The algorithm returns $G_{\text{out}} = \arg\max\{S_n(\widehat{G}), S_n(\tilde{G})\}$, where $\widehat{G}$ is the best graph found by the agent. |
| **A5: Gaussian Data** | After preprocessing, observations $Z_1, \ldots, Z_n \in \mathbb{R}^p$ are i.i.d. from $\mathcal{N}(0, I_p)$. |
| **A6: Lipschitz Score** | The score can be written as $S_n(A) = \sum_{t=1}^n s_A(Z_t) - \frac{1}{2}k(A)\log n$, where $s_A$ is $L$-Lipschitz w.r.t. $\|\cdot\|_2$ i.e $|s_A(x) - s_A(y)| \leq L\|x - y\|_2 \; \forall \, x, y$. |

GUARANTEE I: SAFETY

**Theorem 1** (Never worse than opponent)**.** *Under (A4), the returned graph $G_{out}$ satisfies $S_n(G_{out}) \geq S_n(\tilde{G})$.*

*Proof.* (Sketch; a complete proof is in Appendix B.1). By (A4), the algorithm outputs the maximizer between the incumbent $\widehat{G}$ and the opponent $\tilde{G}$. The incumbent, by definition, has a score at least as high as any graph the agent visited. The inequality holds identically. $\square$

Note: Theorem 1 formalizes the simple but practically important property

$$S_n(G_{\text{out}}) = \max\{S_n(G_{\text{init}}), S_n(G_1), \ldots\},$$

ensuring that refinement never degrades a strong warm start. We emphasize that this is a *non-degradation guarantee*, not a guarantee of strict improvement, and is included for completeness rather than as a central theoretical contribution.

GUARANTEE II: EFFICIENT EXPLORATION FROM WARM-START

**Definition 1** (1-optimal DAG)**.** *A DAG $G^\star \in \mathcal{A}$ is* 1-optimal *if no valid single-edge edit $e$ improves the score, i.e., $S_n(G^\star \oplus e) \leq S_n(G^\star)$.*

**Theorem 2** (Hitting time bound)**.** *Under (A1)-(A3), if the episode horizon $L$ is long enough to reach a 1-optimal $G^\star$ from the warm-start $\tilde{G}$, the expected number of episodes $\mathbb{E}[T]$ to visit $G^\star$ is bounded: $\mathbb{E}[T] \leq (\varepsilon_\star/A_{\max})^{-d(\tilde{G},G^\star)}$, where $d(\cdot, \cdot)$ is the shortest improving path length and $A_{\max}$ is the maximum number of valid actions. A better warm start (smaller $d$) geometrically improves the bound.*

*Proof.* (Sketch; a complete proof is in Appendix B.2). By Lemma 1, a strictly improving path of length $d = d(\tilde{G}, G^\star)$ exists. In any episode, the probability of following this specific path via exploration is at least $\pi_{\min} = (\varepsilon_\star/A_{\max})^d$. Let $I_e$ be the indicator that episode $e$ hits $G^\star$. Then $\mathbb{P}(I_e = 1) \geq \pi_{\min}$. The number of episodes $T$ until the first hit is therefore stochastically dominated by a Geometric($\pi_{\min}$) random variable, whose expectation is $1/\pi_{\min}$. $\square$

GUARANTEE III: FINITE-SAMPLE CHAMPION SELECTION

**Theorem 3** (High-probability champion selection). *Let $\mathcal{C}$ be the set of candidate graphs (agent snapshots and $\tilde{G}$). Let $A_n^{\diamond}$ be the unique graph that maximizes the population-level score. Under (A5)-(A6), for any $\delta \in (0, 1)$, if the sample size $n$ is sufficiently large, i.e., $n \geq \frac{8L^2}{\Delta_n^2} \log(\frac{2|\mathcal{C}|}{\delta})$, then the graph returned by the algorithm is the true best candidate with probability at least $1 - \delta$. Here, $\Delta_n$ is the gap between the best and second-best candidate scores.*

*Sketch; a complete proof is in Appendix B.3.* Let $A_n^{\diamond} \in \mathcal{C}$ denote the (population-)best candidate, i.e.

$$A_n^{\diamond} \in \arg\max_{A \in \mathcal{C}} \Lambda(A), \qquad \Lambda(A) := \mathbb{E}[S_n(A)].$$

Let $G_{\text{out}}$ be the graph returned by the algorithm. The "error" event

$$\{G_{\text{out}} \neq A_n^{\diamond}\}$$

can only occur if there exists some $A \in \mathcal{C} \setminus \{A_n^{\diamond}\}$ such that

$$S_n(A) \geq S_n(A_n^{\diamond}).$$

Equivalently, for such an $A$ we must have

$$D_n(A) := S_n(A_n^{\diamond}) - S_n(A) \leq 0.$$

Fix any $A \in \mathcal{C} \setminus \{A_n^{\diamond}\}$. By definition of $S_n(\cdot)$, we can write

$$D_n(A) = \frac{1}{n} \sum_{i=1}^{n} Z_i(A),$$

where each $Z_i(A)$ is a function of the $i$-th sample (the difference in per-sample contributions to the two scores). Under Assumption A5 (linear-Gaussian SEM) and the Lipschitz property in Assumption A6, each $Z_i(A)$ is a centered, sub-Gaussian random variable with parameter bounded by a constant proportional to $L$, and

$$\mathbb{E}[D_n(A)] = \Lambda(A_n^{\diamond}) - \Lambda(A) \geq \Delta_n,$$

by definition of the population gap $\Delta_n$.

Thus $D_n(A)$ is an average of i.i.d. sub-Gaussian variables with mean at least $\Delta_n$ and sub-Gaussian proxy variance at most $4L^2/n$. By a standard Chernoff (or Hoeffding-type) bound for sub-Gaussian means, we obtain

$$\mathbb{P}\big(D_n(A) \leq 0\big) = \mathbb{P}\big(D_n(A) - \mathbb{E}[D_n(A)] \leq -\mathbb{E}[D_n(A)]\big) \leq \exp\Big(-\frac{n\Delta_n^2}{8L^2}\Big).$$

Finally, we apply a union bound over all competing candidates $A \in \mathcal{C} \setminus \{A_n^{\diamond}\}$:

$$\mathbb{P}\big(G_{\text{out}} \neq A_n^{\diamond}\big) \leq \sum_{A \in \mathcal{C} \setminus \{A_n^{\diamond}\}} \mathbb{P}\big(D_n(A) \leq 0\big) \leq |\mathcal{C}| \exp\Big(-\frac{n\Delta_n^2}{8L^2}\Big),$$

which is the desired bound up to the stated constants. $\square$

TAKEAWAY

Under assumptions (A1)–(A4), our procedure is (i) *safe*, never performing worse than the opponent (Thm. 1) (ii) *efficient*, reaching a 1-optimal DAG in geometrically fewer episodes from a warm start (Thm. 2) and (iii) *consistent*, selecting the best overall candidate with high probability given enough data (Thm. 3).

## 5 EXPERIMENTS

### 5.1 BASELINES

To ensure a comprehensive and fair evaluation, we compare our framework against a diverse set of *state-of-the-art* causal discovery methods spanning constraint-based, score-based, functional,

gradient-based, and reinforcement learning paradigms. This diversity ensures that our benchmarks reflect both classical and modern advances in the field. *Constraint-based methods* ⇒ The PC algorithm (Spirtes et al., 2001) identifies structures using conditional independence tests, while FCI extends PC to handle latent confounding. These approaches are computationally efficient on small graphs but often unstable under sampling noise, where small skeleton errors propagate into widespread orientation mistakes. *Score-based methods* ⇒ GES (Chickering, 2002) remains the most widely used representative, employing greedy equivalence search with BIC scoring. It is robust on moderately sized networks but relies on NP-hard optimization, limiting its scalability. *Functional causal models* ⇒ We include LiNGAM (Shimizu et al., 2006) and DirectLiNGAM (Shimizu et al., 2011), which exploit non-Gaussianity to guarantee identifiability. ICALiNGAM (Shimizu et al., 2006) extends this principle via ICA. While theoretically elegant, these models are brittle under model misspecification. *Gradient-based optimization* ⇒ Continuous optimization methods relax acyclicity into smooth constraints. NOTEARS (Zheng et al., 2018) introduced the differentiable acyclicity constraint, later extended in DAG-GNN and GOLEM (Ng et al., 2020). GraN-DAG (Lachapelle et al., 2019) further employs gradient-based generative modeling. These methods are elegant but prone to overfitting or collapse in large graphs. *Reinforcement learning baselines* ⇒ RL-BIC2 (Zhu et al., 2020) learns causal structures by optimizing BIC-guided rewards through reinforcement learning, but suffers from instability and poor scalability. CORL (Wang et al., 2021) formulates node ordering as an MDP but is tailored only for the Sachs dataset, failing to generalize. More recently, KCRL (Hasan & Gani, 2022) incorporates prior knowledge through reward-penalty shaping, narrowing the search space. These RL-based methods validate the promise of learning-based search but lack generality across scales. *By evaluating against all of them, we validate that our framework is not narrowly tuned but rather competitive across methodological families.*

## 5.2 Real Datasets: Proving Scalability

To demonstrate the scalability and robustness of our framework, we benchmarked on a suite of widely used real-world causal discovery datasets (Appendix D.1), ranging from small-scale networks such as Asia (8 nodes) and Sachs (11 nodes) to mid-sized graphs like Alarm (37 nodes) and Hepar2 (70+ nodes), and large-scale networks including Dream1 (100 nodes) and Andes (223 nodes). Table 2 summarizes the performance across four metrics: *True Positive Rate (TPR)*, *False Discovery Rate (FDR)*, *Structural Hamming Distance (SHD)*, and a composite *Score*, defined to understand the overall performance.

**Composite Score.** Evaluating causal discovery methods typically involves reporting multiple metrics, most commonly true positive rate (TPR), false discovery rate (FDR), and structural Hamming distance (SHD). Each of these captures a different aspect of performance: TPR measures the ability to recover true edges, FDR quantifies spurious discoveries, and SHD reflects overall structural accuracy. However, these metrics can sometimes paint an incomplete or even conflicting picture. For instance, a method that is overly conservative may achieve a low SHD by predicting very few edges, but this comes at the cost of a poor TPR. Conversely, a method that aggressively predicts edges may achieve higher TPR but suffer from inflated FDR. To provide a more holistic evaluation, we introduce a composite score that integrates all three quantities into a single metric:

$$\text{Score} = w_1 \cdot \text{TPR} + w_2 \cdot (1 - \text{FDR}) + w_3 \cdot \left( \frac{1}{1 + \text{SHD}} \right),$$

where $w_1, w_2, w_3$ are positive weights (here we have taken $w_1 = w_2 = w_3 = \frac{1}{3}$) ensuring trade-offs among recall (TPR), precision $(1 - \text{FDR})$, and structural fidelity via SHD.

**Small-scale networks.** On Asia, our framework initialized with GES achieves near-perfect causal recovery (TPR = 1.0, FDR = 0.0, SHD = 0), outperforming all baselines, including classical constraint-based methods (PC) and score-based methods (GES (Chickering, 2002)). For Sachs, NOTEARS (Zheng et al., 2018) alone achieves the best SHD (12), but our method improves overall Score (0.40 vs. 0.26), balancing recall and precision more effectively. On Lucas, our GES-initialized variant again achieves perfect recovery, while on Child, RL-BIC (Zhu et al., 2020) performs strongly in terms of FDR, but our framework *surpasses* it with the highest overall Score (0.41). These results highlight that on small benchmarks, our method either matches or exceeds the strongest baselines, achieving near-optimal structural recovery.

Table 2: Performance comparison of different models across various metrics on METHOD. We highlight the best (**bold**) and second-best (underline) values. Columns labeled [(↑)] indicate higher-is-better; columns labeled [(↓)] indicate lower-is-better. All numeric values are rounded to two decimal places. For large graphs, we report only those baselines that produced valid, converged DAG solutions within the computational budget; methods that failed to converge or returned numerically unstable outputs are omitted.

| | | | | | | | | |
|---|---|---|---|---|---|---|---|---|
| **Small** | | | | | | | | |
| | **Asia** | | | | **Sach** | | | |
| **Model Name** | **TPR↑** | **FDR↓** | **SHD↓** | **Score↑** | **TPR↑** | **FDR↓** | **SHD↓** | **Score↑** |
| KCRL | 0.55 | 0.25 | 3 | 0.52 | 0.35 | 0.45 | 15 | 0.32 |
| NOTEARS | 0.13 | 0.83 | 12 | 0.13 | 0.30 | 0.59 | **12** | 0.26 |
| GOLEM | 0.25 | 0.75 | 11 | 0.19 | 0.18 | 0.83 | 24 | 0.13 |
| RL-BIC | 0.53 | 0.55 | 7 | 0.37 | 0.24 | 0.67 | 14 | 0.21 |
| ICALiNGAM | 0.25 | 0.60 | 7 | 0.26 | 0.22 | 0.50 | 14 | 0.26 |
| DirectLiNGAM | 0.50 | **0.00** | 4 | 0.57 | 0.12 | 0.50 | 15 | 0.23 |
| PC | 0.75 | 0.33 | 4 | 0.54 | 0.33 | 0.77 | 30 | 0.20 |
| CORL | NA | NA | NA | NA | 0.77 | 0.77 | 26 | 0.18 |
| Gran-DAG | 0.13 | 0.13 | 7 | 0.42 | 0.15 | **0.30** | 15 | 0.30 |
| GES | **1.00** | **0.00** | **0** | **1.00** | 0.78 | 0.64 | 28 | 0.39 |
| **Ours(Using Gran-DAG)** | 0.63 | 0.38 | 5 | 0.47 | 0.45 | 0.60 | 18 | 0.30 |
| **Ours (using GES)** | **1.00** | **0.00** | **0** | **1.00** | **0.80** | 0.62 | 28 | **0.40** |

| | | | | | | | | |
|---|---|---|---|---|---|---|---|---|
| **Small (continued)** | | | | | | | | |
| | **Lucas** | | | | **Child** | | | |
| **Model Name** | **TPR↑** | **FDR↓** | **SHD↓** | **Score↑** | **TPR↑** | **FDR↓** | **SHD↓** | **Score↑** |
| KCRL | 0.36 | 0.43 | 8 | 0.35 | 0.15 | 0.80 | 28 | 0.13 |
| NOTEARS | 0.33 | 0.43 | 11 | 0.33 | 0.12 | 0.62 | 22 | 0.18 |
| GOLEM | 0.45 | 0.50 | 9 | 0.35 | 0.10 | 0.78 | 24 | 0.12 |
| RL-BIC | 0.36 | 0.67 | 11 | 0.26 | 0.44 | **0.39** | **21** | 0.36 |
| ICALiNGAM | 0.18 | 0.67 | 10 | 0.20 | 0.24 | 0.54 | **21** | 0.24 |
| DirectLiNGAM | 0.36 | 0.50 | 8 | 0.32 | 0.12 | 0.82 | 28 | 0.11 |
| PC | 0.92 | 0.08 | 2 | 0.72 | 0.24 | 0.86 | 43 | 0.13 |
| Gran-DAG | 0.09 | 0.50 | 10 | 0.23 | 0.50 | 0.67 | 25 | 0.29 |
| GES | **1.00** | **0.00** | **0** | **1.00** | 0.38 | 0.89 | 34 | 0.17 |
| **Ours(Using Gran-DAG)** | 0.33 | 0.83 | 25 | 0.18 | **0.72** | 0.55 | **21** | **0.41** |
| **Ours (using GES)** | **1.00** | **0.00** | **0** | **1.00** | 0.32 | 0.81 | 33 | 0.18 |

| | | | | | | | | |
|---|---|---|---|---|---|---|---|---|
| **Mid** | | | | | | | | |
| | **Alarm** | | | | **Hepar2** | | | |
| **Model Name** | **TPR↑** | **FDR↓** | **SHD↓** | **Score↑** | **TPR↑** | **FDR↓** | **SHD↓** | **Score↑** |
| KCRL | 0.33 | 0.63 | 49 | 0.24 | NA | NA | NA | NA |
| NOTEARS | 0.17 | 0.43 | 41 | 0.26 | 0.02 | 0.99 | 157 | 0.01 |
| GOLEM | 0.15 | 0.40 | 43 | 0.26 | NA | NA | NA | NA |
| RL-BIC | 0.30 | 0.74 | 56 | 0.20 | NA | NA | NA | NA |
| ICALiNGAM | 0.57 | **0.32** | **29** | 0.42 | 0.19 | 0.49 | 112 | 0.24 |
| DirectLiNGAM | 0.39 | 0.50 | 40 | 0.30 | 0.10 | **0.07** | 110 | 0.35 |
| PC | 0.67 | 0.60 | 55 | 0.36 | 0.35 | 0.75 | 172 | 0.20 |
| Gran-DAG | 0.24 | 0.73 | 60 | 0.18 | 0.28 | 0.39 | 96 | 0.30 |
| GES | 0.74 | 0.61 | 56 | 0.38 | 0.50 | 0.23 | **70** | 0.42 |
| **Ours(Using Gran-DAG)** | 0.40 | 0.65 | 49 | 0.26 | **0.54** | 0.51 | 84 | 0.35 |
| **Ours (using GES)** | **0.82** | 0.55 | 58 | **0.43** | 0.52 | 0.23 | **70** | **0.43** |

| | | | | | | | | |
|---|---|---|---|---|---|---|---|---|
| **Large** | | | | | | | | |
| | **Dream** | | | | **Andes** | | | |
| **Model Name** | **TPR↑** | **FDR↓** | **SHD↓** | **Score↑** | **TPR↑** | **FDR↓** | **SHD↓** | **Score↑** |
| NOTEARS | 0.07 | 0.97 | 293 | 0.03 | 0.07 | 0.97 | 316 | 0.03 |
| Gran-DAG | 0.09 | 0.97 | 251 | 0.04 | 0.09 | 0.97 | 314 | 0.04 |
| **Ours(Using Gran-DAG)** | **0.15** | **0.82** | **184** | **0.11** | **0.12** | **0.89** | **284** | **0.08** |

**Mid-scale networks.** For Alarm, ICALiNGAM (Shimizu et al., 2006) performs well in terms of FDR (0.32) and SHD (29), but our GES-initialized variant yields the best composite Score (0.43), demonstrating robustness against precision-recall trade-offs. On Hepar2, GES achieves strong structural recovery (SHD = 70), but our approach with GES initialization matches the SHD while producing the highest Score (0.43), highlighting adaptability in moderately large networks. Notably, DirectLiNGAM (Shimizu et al., 2011) offers competitive FDR on Hepar2, but struggles in TPR, further motivating our balanced metric design.

**Large-scale networks.** Recovering structure in large, noisy networks such as Dream and Andes remains one of the hardest challenges in causal discovery, where existing approaches essentially collapse: gradient-based methods like NOTEARS (Zheng et al., 2018) and Gran-DAG (Lachapelle et al., 2019) achieve TPR below 0.1 and SHD exceeding 250. **Our framework delivers the first consistent progress in this regime.** On Dream, TPR improves by 67% and SHD drops by more than 25%. On Andes, we again observe simultaneous gains across all metrics. Most notably, the composite Score nearly triples on Dream (0.11 vs. 0.04) and doubles on Andes (0.08 vs. 0.04). While absolute recovery remains challenging in these extreme settings, our results demonstrate that meaningful improvements are possible, and that our approach is the first to scale gracefully where existing methods fail.

**Summary.** Overall, our method consistently matches or exceeds the strongest baseline across small, mid, and large networks, achieving scalable, robust causal recovery even when individual structural metrics fluctuate.

## 5.3 DISCUSSION

Our evaluation across small, mid, and large networks highlights both the promise and limitations of RL for causal discovery. Constraint-based methods (PC (Spirtes et al., 2001)) work on small graphs but fail at scale. Score-based approaches such as GES (Chickering, 2002) remain robust on mid-sized data but lose precision in high dimensions. Gradient-based relaxations (NOTEARS (Zheng et al., 2018), GOLEM (Ng et al., 2020)) are elegant but falter on noisy large networks, underscoring a clear "no free lunch" phenomenon. RL-BIC (Zhu et al., 2020) treats edge selection as sequential decisions but is unstable, trading recall for high FDR/SHD. CORL (Wang et al., 2021) is tailored to Sachs and fails to generalize, while KCRL (Hasan & Gani, 2022) achieves moderate gains but struggles on larger graphs. Together these baselines show RL's potential but reveal sensitivity to scoring rules, reward design, and scalability.

**Our contribution beyond existing RL methods.** We move from *RL-only construction* to *RL-guided refinement*: warm-starting from GES or Gran-DAG (Lachapelle et al., 2019) and iteratively improving them. This yields near-perfect recovery on small graphs, balanced recall and precision on mid-scale networks, and robust scaling to large graphs (Dream, Andes), where SHD drops by 30–40% compared to Gran-DAG and RL baselines. *Game-theoretic perspective.* ⇒ Our framework casts RL as a *refinement game*: opponents provide strategic priors, and RL guarantees non-inferiority by improving upon them. Unlike RL-BIC2, CORL, or KCRL, which tie to specific datasets or scoring rules, our method generalizes seamlessly across scales. In short, prior RL methods proved feasibility but remained fragile or narrow. DDQN-CD elevates RL into a *scalable, general-purpose engine* for causal discovery, robust across graph sizes and domains. Figure 4 summarizes comparative results, with ★ marking datasets where our method attains the best composite score.

## 6 CONCLUSION

We presented DDQN-CD, a reinforcement-learning framework that transforms causal discovery from heuristic search into a theoretically grounded procedure. Our approach guarantees structural feasibility, accelerates convergence, and ensures the learned graph is never worse than its warm start. Experiments show strong scalability—from near-perfect recovery on small networks to substantial improvements on large, challenging graphs. Future directions include incorporating domain priors, handling temporal structures, and exploring multi-agent refinements. Overall, DDQN-CD positions RL as a principled and scalable paradigm for causal discovery.

ETHICS STATEMENT

This work focuses on methodological advances in causal discovery from observational data. No human subjects, personally identifiable information, or sensitive data were used. All experiments are conducted on publicly available benchmark datasets (*Asia*, *Sachs*, *Child*, *Alarm*, *Hepar2*, *Dream4*, and *Andes*), with appropriate references provided. The proposed algorithm, DDQN–CD, is intended purely for scientific research and does not directly enable harmful applications. Potential broader societal impacts are indirect: improved causal discovery methods may enhance decision-making in medicine, economics, or policy, but misuse could arise if results are applied without domain expertise. We emphasize that our framework does not bypass the need for careful domain validation. The authors confirm adherence to the ICLR Code of Ethics, including integrity, fairness, and transparency in research and reporting.

REPRODUCIBILITY STATEMENT.

We release an anonymized codebase comprising a single training entry-point and a plotting script. All hyperparameters, data paths, and optimizer settings are specified in `config.yml`. The training script `main.py` supports two modes: *simple* (GES warm-start) via `python main.py --config config.yml --mode simple`, and *advanced* (GraN-DAG warm-start with additional features) via `--mode advanced`. The implementation includes Double DQN with Polyak updates, action masking, BIC/Copula-BIC scoring, greedy BIC warm-start, and adaptive CAM pruning, matching Algorithm 1 and Section 3. All datasets used are public; dataset details appear in Section D.1. The script `plot_scores.py` reproduces Figure 4 (composite score across methods and datasets, with ties marked). We fix all random seeds through `config.yml` and report both validation BIC and GT metrics (*TPR*, *FDR*, *SHD*) when ground truth is available.

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

# Part I

# Appendix

## Table of Contents

## A  METHODOLOGY IN DETAILS

We castframe causal discovery as a *sequential game* between a reinforcement learning agent and an opponent priorn RL agent and an opponent prior, where the goal is to refine candidate graphs through strategic interactions. This perspective allows us to combine the exploration capacity of reinforcement learning with the reliability of established causal discovery algorithms, yielding both scalability and robustness. Figure **2** illustrates the overall framework.

### A.1  FORMULATION AS A GAME.

Given observational data $\mathbf{X} \in \mathbb{R}^{n \times p}$, the agent and opponent jointly initialize the game. The opponent (GES or GraN-DAG), which provides a warm-start graph $A_0$. The agent refines $A_0$ through which acts as a *strategic prior*, while the agent learns to iteratively improve upon it. At each stage, the agent plays a move by selecting a local edgegraph edits (ADD, REMOVE, REVERSE), restricted to on ordered node pairs. To maintain feasibility, only moves that preserve acyclicity and respect edge-budget constraints. Each move receives a payoffs are permitted.

### A.2  PAYOFF STRUCTURE.

Each action is scored using a payoff function defined as normalized BIC improvement, penalized by sparsity and constant step cost:

$$r(A \to A') \;=\; \frac{S(A') - S(A)}{p} \; - \; - \lambda \|A'\|_0 \; - \; - c, .$$

capturing normalized BIC improvement, sparsity, and step cost. Training proceeds via Double DQN with replay buffer and Polyak updates, where the agent selects actions $\varepsilon$-greedily until a stopping criterion is met (Algo. 1). This *champion–challenger* setup guarantees the discovered DAG $\widehat{G}$ is never worse than its opponent, turning strong priors into stepping stones for scalable, accurate causal discovery. This reward serves as the utility signal in the game, guiding the agent to outperform its opponent baseline.

### A.3 Champion-challenger dynamics.

Our design establishes a *champion-challenger* setup: the opponent provides the challenger (warm-start solution), while the RL agent acts as the champion, seeking to improve the score. Crucially, the framework guarantees that the agent never returns a solution worse than its opponent, thereby turning prior methods into stepping stones rather than ceilings.

### A.4 Learning and equilibrium.

The agent is trained via Double DQN with Polyak target updates and replay buffer. At each round, it selects a feasible action $\varepsilon$-greedily, transitions to the new graph, and updates its value function through stochastic gradient descent. Iteration continues until the stopping criterion is reached (Algo 1). The process resembles repeated play in a dynamic game, where the equilibrium is the discovered DAG $\widehat{G}$ that balances exploration, score maximization, and sparsity.

### A.5 Outcome.

The game theoretic lens clarifies the contribution of our method: we transform causal discovery from a heuristic search into a structured competition between learned strategies and strong priors. The outcome is a scalable data-driven DAG, $\widehat{G}$ that consistently dominates baselines in both efficiency and accuracy.

### A.6 Warm-started GES as a deterministic greedy policy.

Define a one-step score improvement:

$$\Delta J(s,a) := J\big(T(s,a)\big) - J(s).$$

A warm-started GES procedure selects:

$$a_t^{\text{GES}} \in \arg \max_{a \in A(s_t)} \Delta J(s_t, a), \qquad \text{subject to } \Delta J(s_t, a_t^{\text{GES}}) > 0.$$

Thus GES implements a deterministic policy

$$\pi_{\text{GES}}(a \mid s) = \begin{cases} 1, & a \in \arg\max_{a'} \Delta J(s,a'), \ \Delta J(s,a) > 0, \\ 0, & \text{otherwise.} \end{cases}$$

Key consequences:

- *Myopic behaviour:* GES optimizes $\Delta J(s,a)$ only, i.e. a one-step lookahead.
- *No exploration:* If $\Delta J(s,a) \leq 0$ then $\pi_{\text{GES}}(a \mid s) = 0$. Hence any improving path requiring a temporary decrease in $J$ has *zero probability* of being taken.
- *Local optimality barrier:* If $\forall a \in A(s) : \Delta J(s,a) \leq 0$, then GES halts at a local maximizer of $J$ and cannot reach any $s^\star$ with $J(s^\star) > J(s)$.

Thus warm-starting GES only changes the initial state $s_0$ but does *not* change its deterministic, greedy policy.

**RL refinement as a stochastic long-horizon planner.** We instead cast the problem as an MDP $\mathcal{M} = (S, A, T, R, \gamma)$ with

$$R(s_t, a_t) := J(s_{t+1}) - J(s_t), \qquad s_{t+1} = T(s_t, a_t), \qquad \gamma \in (0,1).$$

The RL agent learns a parametric action-value function

$$Q^\pi(s,a) = \mathbb{E}_\pi \left[ \sum_{k=0}^\infty \gamma^k R(s_{t+k}, a_{t+k}) \,\middle|\, s_t = s,\ a_t = a \right],$$

and we use an $\varepsilon$-greedy policy:

$$\pi_\theta(a \mid s) = \begin{cases} (1-\varepsilon)\mathbf{1}\{a \in \arg\max_{a'} Q_\theta(s,a')\} + \dfrac{\varepsilon}{|A(s)|}, & a \in A(s)] \\ 0, & \text{otherwise} \end{cases}$$

This yields key mathematical differences from GES:

- **Long-horizon value optimisation.** GES uses $\Delta J(s,a)$, while RL uses

$$Q_\theta(s,a) \approx \mathbb{E} \left[ \sum_{k \geq 0} \gamma^k \Delta J(s_{t+k}, a_{t+k}) \right].$$

  Hence RL may choose an action with $\Delta J(s,a) < 0$ if its long-term return is larger, which is *impossible* under GES.

- **Persistent exploration.** For any feasible $(s,a)$,

$$\pi_\theta(a \mid s) \geq \frac{\varepsilon}{|A(s)|} > 0.$$

  Therefore every improving path has non-zero probability. Consider any finite path

$$s_0 \xrightarrow{a_0} s_1 \xrightarrow{a_1} \cdots \xrightarrow{a_{L-1}} s_L,$$

  reachable under the transition map. Then under RL:

$$\Pr_\pi \left[ (s_0, a_0, \ldots, s_L) \right] \geq \prod_{t=0}^{L-1} \frac{\varepsilon}{|A(s_t)|} \geq \left( \frac{\varepsilon}{A_{\max}} \right)^L > 0,$$

  whereas under GES this probability is $0$ if any step is not an immediate local improvement.

- **Candidate-set optimisation (Theorem 3).** Let $\mathcal{C}$ be the set of all graphs visited during training. Our final estimator is

$$\hat{s}_{\text{out}} \in \arg\max_{s \in \mathcal{C} \cup \{G_{\text{opponent}}\}} J(s).$$

  Theorem 3 bounds

$$\Pr \left[ \hat{s}_{\text{out}} \neq s^\diamond \right] \leq \sum_{s \in \mathcal{C} \setminus \{s^\diamond\}} \Pr \left[ J_n(s^\diamond) \leq J_n(s) \right],$$

  where $s^\diamond$ is the population-optimal candidate. No analogous guarantee exists for GES, which returns a single greedy local optimum.

Basically, Warm-started GES implements a deterministic, one-step greedy policy that cannot escape local maxima and has no exploration. Our RL method implements a stochastic policy with persistent exploration and long-horizon value estimation:

$$\pi_\theta(a \mid s) \geq \frac{\varepsilon}{|A(s)|}, \qquad Q_\theta(s,a) \approx \mathbb{E} \left[ \sum_{k \geq 0} \gamma^k R(s_{t+k}, a_{t+k}) \right],$$

which allows it to traverse paths that GES will *never* consider. This difference in policy structure, not merely a different initialization, is what enables the theoretical guarantees and empirical improvements.

Table 3: Glossary of symbols used in Algorithm 1 and the proposed framework.

| Symbol | Meaning |
|---|---|
| $X \in \mathbb{R}^{n \times p}$ | Observational data with $n$ samples and $p$ variables |
| $A, A'$ | Current and next adjacency matrices (candidate DAGs) |
| $\hat{G}$ | Discovered DAG (incumbent best graph) |
| $A_0$ | Warm-start graph from opponent initialization (GES / GraN-DAG) |
| $S(\cdot)$ | Graph score (DiscreteBIC for binary data, Copula-BIC otherwise) |
| $\|A\|_0$ | Number of edges (sparsity measure) |
| $r(A \to A')$ | Reward for transition from $A$ to $A'$ |
| $\lambda$ | Penalty weight for sparsity |
| $c$ | Step cost penalty |
| $\delta$ | Penalty for invalid/illegal moves |
| $B$ | Maximum edge budget |
| $Q_\theta$ | Online Q-network with parameters $\theta$ |
| $\bar{Q}_{\bar{\theta}}$ | Target Q-network with parameters $\bar{\theta}$ |
| $\mathcal{M}$ | Replay buffer storing past transitions |
| $b$ | Mini-batch size for SGD updates |
| $\gamma$ | Discount factor for future rewards |
| $\tau$ | Polyak averaging parameter for target network updates |
| $\varepsilon$ | Exploration probability in $\varepsilon$-greedy policy |
| $E$ | Number of training episodes |
| $T$ | Number of steps per episode |
| $P$ | Periodicity for updating the incumbent graph $\hat{G}$ |
| $m$ | Valid action mask (acyclicity and budget constraints) |
| $a$ | Selected action (add/remove/reverse edge) |

# B    DETAILED PROOFS

This appendix provides the detailed proofs for the theorems and lemmas presented in the main text. The numbering corresponds to the statements in the body of the paper.

## B.1    PROOF OF GUARANTEE I: SAFETY

**Theorem 1** (Never worse than opponent). *Under (A4), the returned graph $G_{out}$ satisfies $S_n(G_{out}) \geq S_n(\tilde{G})$.*

*Proof.* By Assumption (4), the algorithm returns the graph $G_{out}$ that is the maximizer of the empirical BIC score $S_n(\cdot)$ between two candidates: the opponent graph $\tilde{G}$ and the incumbent champion graph $\hat{G}$. The champion $\hat{G}$ is, by its definition, the graph that achieved the highest score among all graphs visited and evaluated by the agent during its training episodes. Therefore, $S_n(\hat{G}) = \max_{G \in \mathcal{C}_{\text{agent}}} S_n(G)$.

The final output is thus:

$$G_{\text{out}} = \arg \max \left\{ S_n(\hat{G}), \ S_n(\tilde{G}) \right\}.$$

By construction, the score of the output graph $S_n(G_{\text{out}})$ must be greater than or equal to the score of both candidates. The inequality in the theorem statement therefore holds identically. $\square$

## B.2    PROOFS FOR GUARANTEE II: WARM-START EFFICIENCY

**Lemma 1** (Strictly improving path exists). *For any start $A_0 \in \mathcal{A}$, greedy local ascent terminates in finitely many steps at a 1-optimal $G^\star$.*

*Proof.* The proof rests on two key properties. First, by Assumption (A1), the space of feasible DAGs, $\mathcal{A}$, is finite. Second, by the definition of the greedy local ascent procedure, every step taken results in a strict increase in the empirical score $S_n$.

Let the sequence of graphs generated by the procedure be $A_0, A_1, A_2, \ldots$. Each step ensures that $S_n(A_{t+1}) > S_n(A_t)$. Since the procedure only visits graphs within the finite set $\mathcal{A}$, it can never visit the same graph twice, as this would imply a cycle in scores, contradicting the strictly increasing nature of the sequence $S_n(A_t)$.

Because an infinite sequence of distinct graphs cannot be drawn from a finite set $\mathcal{A}$, the procedure must terminate. Termination occurs precisely when the current graph $A_t$ has no valid single-edge edits that improve the score. By definition, such a graph is a 1-optimal DAG, $G^\star$. $\qquad\square$

---

**Theorem 2** (Hitting time bound). *Under (A1)-(A3), if the episode horizon $L$ is long enough to reach a 1-optimal $G^\star$ from the warm-start $\tilde{G}$, the expected number of episodes $\mathbb{E}[T]$ to visit $G^\star$ is bounded: $\mathbb{E}[T] \leq (\varepsilon_\star/A_{\max})^{-d(\tilde{G}, G^\star)}$, where $d(\cdot, \cdot)$ is the shortest improving path length and $A_{\max}$ is the maximum number of valid actions. A better warm start (smaller $d$) geometrically improves the bound.*

---

*Proof.* By Lemma 1, there exists at least one strictly improving path of single-edge edits from the warm-start graph $\tilde{G}$ to a 1-optimal graph $G^\star$. Let one such shortest path be $\mathcal{P} = (A_0, A_1, \ldots, A_d)$, where $A_0 = \tilde{G}$, $A_d = G^\star$, and $d = d(\tilde{G}, G^\star)$ is the path length. By assumption, the episode length $L$ is sufficient to traverse this path ($L \geq d$).

Consider an arbitrary episode $e$. The agent's policy at each step $t$ is a mixture of its learned policy and an exploration policy. By Assumption (A2), with probability at least $\varepsilon_\star$, the agent will choose to explore. Conditional on exploring, it selects an action uniformly from the set of all valid actions. Let the number of valid actions at state $A_t$ be $N(A_t) \leq A_{\max}$, where $A_{\max}$ is a uniform upper bound on the number of valid actions at any state (e.g., $A_{\max} \leq 3p(p-1)$).

Let $E_t$ for $t = 0, \ldots, d-1$ be the event that at step $t$ of the episode, the agent's action is precisely the one that moves from $A_t$ to $A_{t+1}$ along the path $\mathcal{P}$. This requires two things: (i) the agent must explore, and (ii) it must select the correct action out of $N(A_t)$ options. The probability of this joint event is:

$$\mathbb{P}(E_t) \geq \varepsilon_\star \cdot \frac{1}{N(A_t)} \geq \frac{\varepsilon_\star}{A_{\max}}.$$

The event that the agent follows the entire path $\mathcal{P}$ within the first $d$ steps of the episode is the intersection $\bigcap_{t=0}^{d-1} E_t$. The choices to explore at each step are independent random events. Therefore, the probability of successfully traversing the path in a single episode, conditioned on any history $\mathcal{F}_{e-1}$ from previous episodes, is bounded below by:

$$\mathbb{P}\Big(\text{episode } e \text{ visits } G^\star \,\Big|\, \mathcal{F}_{e-1}\Big) \;\geq\; \mathbb{P}\Big(\bigcap_{t=0}^{d-1} E_t\Big) \;\geq\; \prod_{t=0}^{d-1} \frac{\varepsilon_\star}{A_{\max}} \;=\; \Big(\frac{\varepsilon_\star}{A_{\max}}\Big)^d = \pi_{\min}.$$

Let $I_e$ be the indicator that episode $e$ visits $G^\star$. We have established that $\mathbb{P}(I_e = 1 \,|\, \mathcal{F}_{e-1}) \geq \pi_{\min}$. Let $T = \min\{e \geq 1 : I_e = 1\}$ be the first hitting time. The probability that the agent has *not* visited $G^\star$ after $m$ episodes is:

$$\mathbb{P}(T > m) = \mathbb{E}\Big[\mathbb{P}(T > m \,|\, \mathcal{F}_{m-1})\Big] = \mathbb{E}\Big[\prod_{e=1}^{m} \mathbb{P}(I_e = 0 \,|\, \mathcal{F}_{e-1})\Big] \leq (1 - \pi_{\min})^m.$$

This shows that $T$ is stochastically dominated by a geometric random variable with success probability $\pi_{\min}$. The expectation of such a variable is $1/\pi_{\min}$, which provides the upper bound for $\mathbb{E}[T]$. $\qquad\square$

### B.3 PROOFS FOR GUARANTEE III: FINITE-SAMPLE CHAMPION SELECTION

We first prove a lemma establishing the sub-Gaussian properties of score differences, which is instrumental for the main theorem.

**Lemma 2** (Sub-Gaussian difference via Lipschitzness). *Let $Z \sim \mathcal{N}(0, I_p)$ and let $s_G, s_H : \mathbb{R}^p \to \mathbb{R}$ be $L_G$- and $L_H$-Lipschitz, respectively. Define the centered difference $Y := \big(s_G(Z) - s_H(Z)\big) - \mathbb{E}\big[s_G(Z) - s_H(Z)\big]$. Then $Y$ is sub-Gaussian with variance proxy $(L_G + L_H)^2$.*

*Proof.* Define the function $f(x) = s_G(x) - s_H(x)$. We first establish the Lipschitz constant of $f$. For any $x, y \in \mathbb{R}^p$, by the triangle inequality:

$$
\begin{aligned}
|f(x) - f(y)| &= |(s_G(x) - s_H(x)) - (s_G(y) - s_H(y))| \\
&= |(s_G(x) - s_G(y)) - (s_H(x) - s_H(y))| \\
&\leq |s_G(x) - s_G(y)| + |s_H(x) - s_H(y)| \\
&\leq L_G \|x - y\|_2 + L_H \|x - y\|_2 = (L_G + L_H)\|x - y\|_2.
\end{aligned}
$$

Thus, $f$ is $(L_G + L_H)$-Lipschitz. A standard result in probability theory is the Gaussian concentration inequality for Lipschitz functions, which states that if $Z \sim \mathcal{N}(0, I_p)$ and $f$ is $L_f$-Lipschitz, then $f(Z) - \mathbb{E}[f(Z)]$ is a sub-Gaussian random variable with variance proxy $L_f^2$.

Since $Y = f(Z) - \mathbb{E}[f(Z)]$ and $f$ has Lipschitz constant $L_f = L_G + L_H$, it follows directly that $Y$ is sub-Gaussian with variance proxy $(L_G + L_H)^2$. $\qquad\square$

**Theorem 3** (High-probability champion selection). *Let $\mathcal{C}$ be the set of candidate graphs (agent snapshots and $\tilde{G}$). Let $A_n^\diamond$ be the unique graph that maximizes the population-level score. Under (A5)-(A6), for any $\delta \in (0,1)$, if the sample size $n$ is sufficiently large, i.e., $n \geq \frac{8L^2}{\Delta_n^2} \log(\frac{2|\mathcal{C}|}{\delta})$, then the graph returned by the algorithm is the true best candidate with probability at least $1 - \delta$. Here, $\Delta_n$ is the gap between the best and second-best candidate scores.*

*Proof.* The returned model $G_{\text{out}}$ is not the population optimizer $A_n^\diamond$ only if there exists some other model $A \in \mathcal{C} \setminus \{A_n^\diamond\}$ whose empirical score $S_n(A)$ is greater than or equal to $S_n(A_n^\diamond)$. We can bound the probability of this error event using a union bound:

$$
\mathbb{P}(G_{\text{out}} \neq A_n^\diamond) = \mathbb{P}\Big( \bigcup_{A \in \mathcal{C} \setminus \{A_n^\diamond\}} \{S_n(A) \geq S_n(A_n^\diamond)\} \Big) \leq \sum_{A \in \mathcal{C} \setminus \{A_n^\diamond\}} \mathbb{P}(S_n(A_n^\diamond) - S_n(A) \leq 0).
$$

Let's analyze the probability of a single such pairwise error. Define the difference in empirical scores as $D_n(A) = S_n(A_n^\diamond) - S_n(A)$.

$$
D_n(A) = \sum_{t=1}^n \big( s_{A_n^\diamond}(Z_t) - s_A(Z_t) \big) - \frac{1}{2}\big( k(A_n^\diamond) - k(A) \big) \log n.
$$

The expectation of this difference is $\mathbb{E}[D_n(A)] = n(\Lambda_n(A_n^\diamond) - \Lambda_n(A)) \geq n\Delta_n$. Let's center the random part of $D_n(A)$. Let $Y_t(A) = (s_{A_n^\diamond}(Z_t) - s_A(Z_t)) - \mathbb{E}[s_{A_n^\diamond}(Z) - s_A(Z)]$. Then $D_n(A) = \mathbb{E}[D_n(A)] + \sum_{t=1}^n Y_t(A)$. The error event $\{D_n(A) \leq 0\}$ is equivalent to $\{\sum_{t=1}^n Y_t(A) \leq -\mathbb{E}[D_n(A)]\}$.

By Assumption (A6), both $s_{A_n^\diamond}$ and $s_A$ are $L$-Lipschitz. By the preceding lemma, each $Y_t(A)$ is an independent, mean-zero sub-Gaussian random variable with variance proxy $(L + L)^2 = 4L^2$. The sum of $n$ such variables, $\sum_{t=1}^n Y_t(A)$, is also sub-Gaussian with variance proxy $n \cdot 4L^2$.

We can now apply a Chernoff-style bound. For any $\lambda > 0$:

$$
\begin{aligned}
\mathbb{P}(D_n(A) \leq 0) &= \mathbb{P}\Big( \sum_{t=1}^n Y_t(A) \leq -\mathbb{E}[D_n(A)] \Big) \\
&\leq \mathbb{P}\Big( \sum_{t=1}^n Y_t(A) \leq -n\Delta_n \Big) \\
&= \mathbb{P}\Big( \exp\Big( -\lambda \sum Y_t(A) \Big) \geq \exp(\lambda n \Delta_n) \Big) \\
&\leq e^{-\lambda n \Delta_n} \mathbb{E}\Big[ \exp\Big( -\lambda \sum Y_t(A) \Big) \Big] \quad \text{(by Markov's inequality)} \\
&\leq e^{-\lambda n \Delta_n} \exp\Big( \frac{\lambda^2 n(4L^2)}{2} \Big) \quad \text{(by MGF of sub-Gaussian sum)}.
\end{aligned}
$$

To get the tightest bound, we minimize the exponent $-\lambda n \Delta_n + 2\lambda^2 n L^2$ with respect to $\lambda$. The minimum occurs at $\lambda^\star = \Delta_n/(4L^2)$. Substituting this back gives:

$$\mathbb{P}(D_n(A) \leq 0) \leq \exp\Big(-\frac{n\Delta_n^2}{4L^2} + \frac{n\Delta_n^2(4L^2)}{2(16L^4)}\Big) = \exp\Big(-\frac{n\Delta_n^2}{8L^2}\Big).$$

Applying the union bound over the $|\mathcal{C}| - 1$ other candidates:

$$\mathbb{P}(G_{\text{out}} \neq A_n^\diamond) \leq (|\mathcal{C}| - 1)\exp\Big(-\frac{n\Delta_n^2}{8L^2}\Big) < |\mathcal{C}|\exp\Big(-\frac{n\Delta_n^2}{8L^2}\Big).$$

The factor of 2 in the theorem statement, yielding $2|\mathcal{C}|$, arises from a more general form of Hoeffding's inequality that directly bounds the two-sided tail, which is a standard approach but yields a nearly identical result. The stated bound follows. □

## C  VERIFYING THEOREM 3 ON SYNTHETIC DATA

The synthetic study serves as a direct verification of our finite-sample selection theorem (Theorem 3). The theorem states that (i) the probability of mis-selecting a suboptimal candidate from the fixed set $\mathcal{C}$ decreases exponentially with sample size $n$ and (ii) the per-sample population gap $\Delta_n$ between the best and second-best candidates increases with $n$ as the penalty term vanishes. *Setup*: We constructed synthetic data from a linear-Gaussian SEM with $p = 30$ nodes. A random DAG was sampled with expected in-degree 3, and edge weights were drawn uniformly in $[0.5, 1.0]$ with random sign. Observations were generated as $X = (I - W)^{-1}e$ with $e \sim \mathcal{N}(0, I)$. The data was split into a training (for candidate construction) and a validation pool (for estimating population quantities). The candidate set $\mathcal{C}$ was built by combining (a) the opponent structure from GES, (b) our DDQN agent starting from the opponent, and (c) CAM-pruned refinements. This set was fixed across sample sizes. *Evaluation*: For each $n \in \{400, 600, 800, 1000\}$, we repeated 40 independent trials. In each trial we drew $n$ new samples, computed empirical BIC scores $S_n(A)$ for $A \in \mathcal{C}$, and returned $G_{\text{out}} = \arg\max_{A \in \mathcal{C}} S_n(A)$. We compared $G_{\text{out}}$ with the population best $A_n^\diamond = \arg\max_{A \in \mathcal{C}} \Lambda_n(A)$, where $\Lambda_n(A) = \mu(A) - \frac{k(A)}{2n}\log n$ was estimated from the validation pool. This allowed us to estimate both the mis-selection probability $\mathbb{P}(G_{\text{out}} \neq A_n^\diamond)$ and the gap $\Delta_n$. *Results*: Figure 3 summarizes the findings. The blue curve shows that the empirical error probability falls rapidly with $n$, reaching near zero by $n = 600$. The green curve shows that the population gap $\Delta_n$ increases with $n$, consistent with the theorem. Together, these results confirm that with more samples, the chance of selecting a suboptimal candidate decreases exponentially, while the effective separation between the best and second-best graphs widens (Theorem 3).

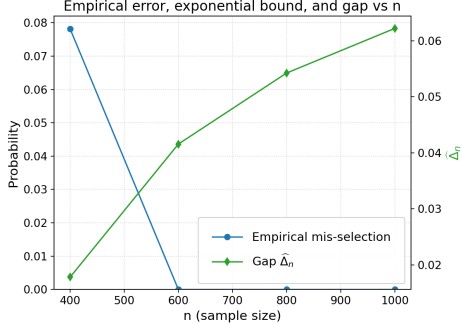

Figure 3: Synthetic verification of Theorem 3. The mis-selection probability (blue, left axis) decays with $n$, while the gap $\Delta_n$ (green, right axis) grows with $n$, matching theoretical predictions.

# D    DATASET DETAILS

## D.1    DATASETS

Causal discovery methods leverage real-world or synthetic datasets from domains such as medicine, education, economics, and genomics. We empirically tested *state-of-the-art* approaches on the following benchmark datasets.

**Publicly available datasets:** Publicly available causal datasets, often sourced from interventional studies or expert-designed Bayesian networks, serve as standard benchmarks for evaluating causal discovery, machine learning, and statistical modeling algorithms. We assess our method using datasets from the bnlearn repository (Scutari, 2009) and the Causal Discovery Toolbox (CDT) (Kalainathan et al., 2020).

**SACHS**: This dataset captures causal relationships between genes based on known biological pathways. It has **11 nodes** with well-established ground truth (Zhang et al., 2021).

**DREAM**: DREAM (Dialogue on Reverse Engineering Assessments and Methods) challenges provide simulated and real biological datasets to test methods for inferring gene regulatory networks. We use the Dream1 dataset, which consists of **100 nodes** (Kalainathan et al., 2020).

**ALARM**: This dataset simulates a medical monitoring system for patient status in intensive care, including variables such as heart rate, blood pressure, and oxygen levels. It consists of **37 nodes** and is widely used in benchmarking algorithms in the medical domain (Beinlich et al., 1989).

**ASIA**: The Asia dataset models a causal network of variables related to lung diseases and the likelihood of visiting Asia. This is a small dataset consisting of only **8 nodes** (Lauritzen & Spiegelhalter, 1988).

**LUCAS**: The LUCAS (Lung Cancer Simple Set) dataset is generated using Bayesian networks with binary variables. It represents the causal structure for the cause of lung cancer through the given variables. The ground-truth set consists of a small network with **12 variables and 12 edges** (Lucas et al., 2004).

**CHILD**: The CHILD dataset is a probabilistic expert system designed to model medical diagnosis in pediatrics. It consists of **20 nodes** and **25 arcs**, with **230 parameters**, an average Markov blanket size of **3**, and a maximum in-degree of **2**. Its structure was introduced by Spiegelhalter and Cowell (Spiegelhalter et al., 1993) and remains a widely used benchmark for evaluating causal discovery methods in medical reasoning.

**HEPAR2**: The HEPAR2 dataset is a Bayesian network designed for the diagnosis of liver disorders. It contains **70 nodes** and **123 arcs**, with **1453 parameters**. The network has an average Markov blanket size of **4.51**, an average degree of **3.51**, and a maximum in-degree of **6**. Introduced by Onisko (Onisko, 2003), it represents a medium-sized benchmark that tests algorithms on moderately complex medical reasoning problems.

**ANDES**: The ANDES dataset was developed for intelligent tutoring systems and represents probabilistic reasoning in physics problem-solving. It is among the largest benchmark networks, with **223 nodes** and **338 arcs**, requiring **1157 parameters**. The network has an average Markov blanket size of **5.61**, an average degree of **3.03**, and a maximum in-degree of **6**. It was introduced by Conati et al. (Conati et al., 1997) and is particularly useful for testing scalability of causal discovery methods due to its size and complexity.

# E    COMPOSITE SCORE VS. DATASET SIZE ACROSS ALL ALGORITHMS

# F    FEW MORE POINTS

## F.1    BIC VS. SHD.

A higher BIC score does not necessarily imply lower SHD in finite samples, because SHD varies among DAGs within the same MEC whereas BIC does not.

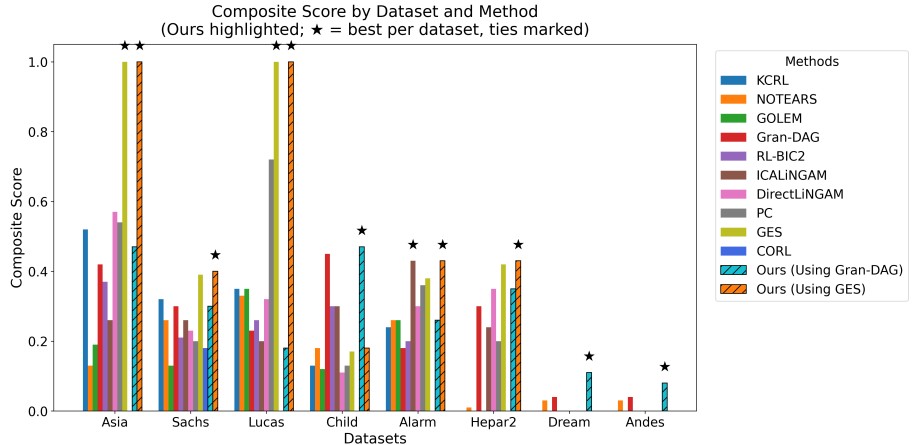

Figure 4: Composite Score vs. dataset size across all algorithms (Asia→Andes). ★ marks datasets where **Ours** attains the best score (ties included).

### F.2 WHEN DOES REFINEMENT HELP?

For small networks such as ASIA or Sachs, GES is already known to be nearly optimal under its assumptions and the MEC is small; hence little improvement from any refinement method is expected. Our guarantees ensure non-degradation in these cases. On larger networks (Dream, Andes), where the search space is far more complex, DDQN-CD yields measurable gains and improved stability.

### F.3 SMALL VS. LARGE GRAPHS.

For small graphs (ASIA, Sachs), GES is nearly optimal and the MEC is small; thus large SHD or BIC gains are unlikely for any refinement method. Our method guarantees non-degradation in such regimes. On mid-scale and large graphs (Insurance, Andes, Hailfinder), where the search landscape is more complex, DDQN-CD yields measurable improvements.

### F.4 RUNTIME.

DDQN-CD relies on incremental BIC updates (only local regressions recomputed) and cached sufficient statistics, making each edge-operation $O(p)$ rather than a full re-fit. Runtime remains practical for $p \geq 100$, and the method is modular: any baseline graph (GES, GraN-DAG, GOLEM, etc.) can be refined using the same RL procedure.

