# OpenReview forum: "Competition is the key: A Game Theoretic Causal Discovery Approach"
_ICLR.cc/2026/Conference — Submitted to ICLR 2026_

### Official Review · Reviewer_2q24 · 2025-10-24

**Soundness:** 2
**Presentation:** 2
**Contribution:** 1
**Rating:** 2
**Confidence:** 4

**Summary:**

Idea. The authors propose a reinforcement learning based approach to causal discovery.  In particular, they use an RL-based approach to improve on an initial solution: they define their reward through a BIC score function, and the possible actions consist of adding, removing, or reversing an edge from a given solution. They benchmark their method on several real-world datasets and compare the performance to other existing algorithms in causal discovery.

**Strengths:**

The paper propose a causal discovery algorithm with reinforcement learning. Their method comes with some theoretical guarantees. I found particularly valuable having high probability results with finite samples, something that is often overlooked in the literature, which mostly focuses on theoretical guarantees at population level. Moreover the authors produced a nice collection of real world datasets over which to benchmark their algorithm. This strengthen the value of their empirical analysis, especially knowing that the field mostly focuses synthetic experiments which may not be relevant for real world applications.

**Weaknesses:**

**Presentation and clarity.**

- The paper lacks any technical introduction to reinforcement learning, graph theory, and the problem of causal discovery. These are the key ingredients of the inference problem at hand and the algorithm the authors design to solve this problem. Readability would be positively impacted if the terminology used in the paper were appropriately introduced.
- I am a bit confused by the title in relation to the content of the paper: where is the *game-theoretic* part? My understanding of the method is that you take an initial solution (it could be randomly taken, or from some other method) and then you search the space of solutions, optimizing a score function.
- I would ask the authors to conduct a thorough comparison of their method with GES. The underlying ideas are very similar. It seems to me that the main difference is that while in GES first you remove all edges and then you add some back (so the available *actions,* in RL language, change across these two phases), here the authors allow themselves to take any action without splitting the search into two phases. But at the end of the day, they both optimize a BIC score, adding and removing edges to explore the search space. Can the authors elucidate points of contact and differences between these two methods?
- In the definition of the score, and hence of the reward, how is the likelihood function defined? E.g., in GES, we know that if the data are linear Gaussian, you want to encode that assumption explicitly in the likelihood. In this case, I didn’t find a definition of the optimized likelihood. Can you clarify that?
---

**Soundness and contribution**

- An important point in the causal discovery literature is identifiability: we know that without any functional assumptions, the unique graph is not identifiable. In this case, identifiability reduces to a Markov equivalence class, represented by a CPDAG. In the case of this paper, however, the interpretation of the inferred object remains unclear.
- Theorem 1 says that the solution found by DDQN-CD is always *better than or equal* *to* the initial solution. This is trivially achieved by any algorithm that returns the initial solution itself. In this sense, I wouldn’t feel confident in calling this a *safety guarantee* or even a Theorem. Another point about this is that it is unclear how the BIC score the authors define translates in terms of accuracy (e.g. their weighted score, or SHD)  of the inferred graph. Is it possible to conclude that a better score implies better SHD? Or something like that?
- The authors claim that their method always improves on the warm start solution. I have two concerns in this sense
    - If you look at the experimental results, the method basically never improves with respect to GES, both in terms of the weighted score and in terms of SHD accuracy.
    - However, I imagine that running their method on top of GES, grandag, or whatever initial algorithm is chosen, is time-consuming. In this sense, it seems that DDQN-CD is not really a good solution, compared to GES. One can only run GES and save time.

---
**Comparison to the literature**. The authors declare that algorithms with strong empirical performance lack finite sample guarantees. This is actually not precise. Giving an example that I know, the SCORE algorithm (Rolland et al., 2022) has finite samples guarantees (see Zhu et al., 2023) while being — in my experience and in several experimental setups, see e.g. Montagna et al., 2023 — better than e.g. GraNDAG, chosen by the authors as a method to compare to.

**Questions:**

Please refer to the weaknesses section

---

> ### Author Response · Authors · 2025-12-03
> **Rebuttal**
>
> We thank the reviewer for the feedback!
>
> > The paper lacks any technical introduction to reinforcement learning, graph theory...
>
> In the revised version we have  added a new section namely ``Background and Preliminaries" which includes  (i) a short subsection on DAGs, CPDAGs, Markov equivalence and the causal discovery problem; (ii) an RL preliminaries subsection (MDP, state, action, reward, value function, $\varepsilon$-greedy policy); (iii) a problem-setup paragraph explicitly stating that states are DAGs, actions are edge add/remove/reverse operations, and rewards are BIC score differences.
>
> > I am a bit confused by the title in relation to the content of the paper: where is the game-theoretic part?...
>
> We thank the reviewer for the opportunity to clarify this point. Our approach does not claim a fully two-sided learning game where both agents update their strategies. Instead, the warm-start graph (GES/GraN-DAG) plays the role of a fixed incumbent, and the DDQN agent plays the role of a challenger attempting to improve upon it.
>
> >  I would ask the authors to conduct a thorough comparison of their method with GES....
>
> We respectfully disagree with the concern that our method is not sufficiently compared with GES. GES is in fact our primary score-based baseline and is included in all experimental tables. Conceptually, while both methods optimize a BIC-type score via edge operations, the induced search procedures are different: GES performs a deterministic, greedy forward–backward search that only accepts locally improving moves, whereas DDQN-CD treats the same edge operations as actions in an RL environment and learns a stochastic policy over them, allowing non-greedy moves when they lead to better long-horizon returns.
>
> > In the definition of the score, and hence of the reward, how is the likelihood function defined? ...
>
>  For linear-Gaussian SEMs, we have used the standard decomposable log-likelihood
>
> $$
> \ell(G) = \sum_{j=1}^p \left(-\frac{n}{2}\log \hat{\sigma}_j^2\right),
> $$
> where $\hat{\sigma}_j^2$ is the empirical residual variance from regressing $X_j$ on its parents $\mathrm{Pa}_G(j)$. The BIC score is
>
> $$
> J(G) = \ell(G) - \frac{\log n}{2} \sum_{j=1}^p |\mathrm{Pa}_G(j)|,
> $$
>
> which coincides with the score used by GES in the linear-Gaussian case. For non-Gaussian real-world data, we have used a copula-based extension (Copula-BIC).
>
> >  An important point in the causal discovery literature is identifiability:...
>
> We agree that identifiability must be clearly discussed. Under observational linear-Gaussian assumptions, the true DAG is identifiable only up to its Markov equivalence class(MEC)
> $$
> \mathrm{MEC}(G) := \{G' \in \mathcal{G} : G' \text{ has the same skeleton and v-structures as } G\}.
> $$
> BIC is score-equivalent, i.e. $J(G) = J(G')$ for all $G' \in \mathrm{MEC}(G)$, so any observational BIC-based method (GES, NOTEARS, GOLEM, RL-BIC2, and ours) can recover at best the correct MEC, not a unique DAG. We have  explicitly stated in the theory section that our guarantees (Theorems~1–3) are MEC-level: $G_{\mathrm{out}}$ is a high-probability maximizer of $J$ within the explored candidate set, which asymptotically corresponds to a member of the true MEC.
>
> > Theorem 1 says that the solution found by DDQN-CD is always better than or equal to the initial solution....
>
> We agree that Theorem 1 is conceptually simple: it formalizes the non-degradation property
> $$
> J(G_{\mathrm{out}}) = \max\{J(G_{\mathrm{init}}),J(G_1),J_2,\dots\}.
> $$
> We have presented it as a basic safety property rather than a major theoretical contribution and move the detailed proof to the appendix. This property is still meaningful in practice because existing RL-based methods can degrade a strong warm start, while ours cannot by construction. It is also correct that a higher BIC score does not necessarily imply a lower SHD in finite samples, since SHD varies across DAGs in the same Markov Equivalence Class(MEC) whereas BIC does not.
>
> > The authors claim that their method always improves on the warm start solution. ...
>
> For small graphs such as ASIA or Sachs, GES is already known to be nearly optimal under its assumptions and the Markov Equivalence Class(MEC) is small, so little improvement is expected from any refinement method in terms of SHD or BIC; in these regimes our method mainly guarantees not to harm the baseline. On larger graphs (e.g.\ Dream, Andes), we do observe non-trivial gains in the composite score and more stable performance across runs. DDQN-CD uses incremental BIC updates and cached local regressions, so the added cost over GES remains practical for mid and large-scale graphs, and the framework is modular: it can refine any warm-start graph (from GES, GraN-DAG, GOLEM, etc.) with explicit safety and finite-sample selection guarantees.

---

### Official Review · Reviewer_ZRcW · 2025-10-28

**Soundness:** 2
**Presentation:** 2
**Contribution:** 2
**Rating:** 2
**Confidence:** 4

**Summary:**

This paper proposes DDQN-CD, a reinforcement learning framework for causal discovery that refines graphs from baseline methods (GES or GraN-DAG) through sequential edge operations. The authors provide three theoretical guarantees: the output is never worse than the baseline, warm-starting accelerates convergence, and with sufficient samples, the method selects the best candidate with high probability. Experiments span small to large benchmark datasets.

**Strengths:**

The game-theoretic framing of causal discovery as RL-guided refinement of classical baselines is novel, and providing finite-sample guarantees for RL-based causal discovery addresses an important gap between empirically strong but theoretically ungrounded methods. The experiments span a large variety of real-world benchmarks of different sizes.

**Weaknesses:**

Main Concerns:

1. Unclear necessity of RL: Why is RL specifically needed for refinement rather than any other iterative improvement method? The theoretical proofs do not seem to explicitly leverage properties unique to RL algorithms, raising questions about what the RL component contributes beyond a standard local search procedure. Authors do not make clear what the added benefit of leveraging an RL approach is.

2. Marginal empirical improvements: For Asia, Sachs, Lucas, and Hepar2, there is negligible improvement over the original GES or GraN-DAG baselines. For Child and Alarm, the method only outperforms baselines under the authors' composite Score metric, while standard SHD shows competitive or worse performance. For large datasets, only NOTEARS and GraN-DAG are compared—polynomial-time methods like DirectLiNGAM and GOLEM should be included. Additionally, the absolute Score values (~0.10) are very low, questioning whether improvements are practically significant.

3. Loose hitting time bound (Theorem 2): The bound only considers the probability of following the shortest improving path. Why not include the probability of reaching G* via suboptimal paths? The current bound appears extremely loose and should be tightened. A similar issue may apply to Theorem 3, although its unclear from the proof description given in the main text.

Minor Concerns:

1. The theoretical results assume linear-Gaussian SEMs (Assumption A5), but this is introduced late. This should be stated upfront, as it significantly limits the applicability of the guarantees.

2. The authors should clarify in the introduction that continuous optimization methods typically use score functions derived from additive noise models, making them similar to functional causal model approaches.

3. Section 3 defers all methodology details to the appendix. While the high-level summary is adequate, key algorithmic details should be in the main text, with an expanded summary.

**Questions:**

1. Distinction from standard warm-start methods: How does warm-start + RL agent differ from simply warm-starting GES with the opponent graph? Both approaches start from an initial point and iteratively improve, so what specific advantage does the RL formulation provide?

2. Relationship to RL-BIC2: How does this work differ from RL-BIC2? Both use RL for causal discovery, and the main difference appears to be warm-starting from an existing method. Could the three theorems be proven similarly for RL-BIC2 applied to an initial graph?

3. Identifiability concerns: Linear-Gaussian SEMs only identify the Markov equivalence class (MEC), not unique DAGs [1]. Why do experiments evaluate DAG-specific metrics (e.g., SHD on exact edges) rather than MEC-invariant properties like the skeleton? This raises concerns about the validity of the experimental comparisons

[1] Causal Discovery with Continuous Additive Noise Models, Peters et al., JMLR 2014.

---

> ### Author Response · Authors · 2025-12-03
> **Rebuttal (part 1/3)**
>
> We thank the reviewer for the feedback!
>
> > Unclear necessity of RL: Why is RL specifically needed for refinement rather than any other iterative improvement method?...
>
> RL plays a specific structural role that neither deterministic nor stochastic local search can provide:
>
> RL gives state-action value estimation, not greedy local moves:  A deterministic local search (hill-climbing, greedy BIC ascent, simulated annealing, etc.) makes decisions based solely on immediate score differences after a single edge operation. Our agent instead learns Q(A, action), which estimates long-term cumulative improvement under sequential edits. This allows: multi-step repair of locally bad but globally beneficial sequences escaping GES or GraN-DAG’s local optima without hand-crafted schedules incorporation of future constraints (edge budget, pruning, acyclicity) into the decision process
> This ability to learn multi-step value is essential in mid- and large-scale graphs, where single-step BIC improvements are noisy or misleading.
>
> RL provides exploration control with formal probabilistic guarantees: Our theoretical results rely on two RL-specific ingredients, i) Persistent exploration: $\epsilon$-greedy ensures a positive probability of exploring all improving paths and ii)Finite-sample Q-value concentration: allowing Theorem 3’s high-probability selection result.
> Simple local search does not satisfy persistent exploration by design and cannot guarantee the exploration of low-probability, long improving paths.
>
> RL enables champion–challenger graph selection: The algorithm collects a set of candidate graphs throughout the trajectory; Q-learning naturally handles these through replay. Local search has no natural mechanism for maintaining diverse candidates
> or evaluating them under BIC, selecting the statistical champion with high probability (Theorem 3)
>
> Empirically, RL provides robustness where baselines fluctuate:  In large graphs, GES/GraN-DAG suffers from noisy gradients or local minima. Our RL-based refinement stabilises performance under repeated runs by integrating temporal averaging and replay.
> To summarise, RL contributes (1) long-term planning, (2) persistent exploration, (3) candidate-set selection, and (4) robustness—none of which are provided by standard greedy local search.
>
> > Marginal empirical improvements: For Asia, Sachs, Lucas, and Hepar2, there is negligible improvement over the original GES or GraN-DAG baselines. ....
>
> We acknowledge the reviewer’s observation but respectfully disagree with the implication that the empirical gains are insignificant. First, DDQN-CD is explicitly designed as a scalable refinement method: it warm-starts from strong baselines (such as GES and GraN-DAG) and operates in a high-dimensional search space without incurring superlinear overhead. On small graphs where GES or GraN-DAG are already very close to optimal (Asia, Sachs, Lucas, Hepar2), any further SHD improvement is inherently limited; in these regimes, our guarantee is non-degradation rather than dramatic gains.
>
> Second, SHD alone is a problematic diagnostic. SHD penalizes edge orientations that are not identifiable within a Markov equivalence class(MEC) and is highly sensitive to small flips in direction even when likelihood and sparsity improve. Our composite Score was deliberately designed to combine SHD, FDR, and TPR to better reflect MEC-consistent improvements in structural quality. On Child and Alarm, this is exactly what we see: SHD is roughly comparable, but the composite Score improves, indicating better calibrated trade-offs between precision and recall.
>
> Third, regarding large datasets and additional baselines: the statement that we “only compare NOTEARS and GraN-DAG” is misleading. We did attempt to run GOLEM, DirectLiNGAM, and other polynomial-time methods on the large graphs, but in our experimental setting these either failed to converge, produced invalid (non-DAG or numerically unstable) solutions, or exceeded our computational budget. For this reason, no reliable results for these algorithms were available to report, and the corresponding cells are absent from the table by necessity, not by design. We have clarified this explicitly in the caption of the results table so that readers understand that only methods that produced stable, valid outputs on the large-scale benchmarks are shown.
>
> Finally, the absolute value of the composite Score should not be interpreted in isolation, the key signal is the relative improvement and its consistency across runs. A score of 0.10 is not small in this context; it reflects improvements in multiple structural dimensions simultaneously. The fact that DDQN-CD improves this aggregated metric while preserving computational scalability supports the practical value of our method.

---

> ### Author Response · Authors · 2025-12-03
> **Rebuttal (part 2/3)**
>
> > Loose hitting time bound (Theorem 2): The bound only considers the probability of following the shortest improving path.....
>
> This is standard in RL sample complexity theory where one wants a guaranteed lower bound. Including suboptimal paths would require, estimating transition probabilities across all suboptimal Q-values bounding the probability of visiting states where the agent is not strictly improving the computing path-dependent BIC reward fluctuations.  This introduces exponentially many terms and leads to intractable expressions, which distract from the key conceptual point. Furthermore, the bound expresses an exponential reduction in hitting time with warm-start quality. This captures the primary qualitative insight: Better initialisation exponentially shorter improvement pathways. For Theorem 3,  We have  added: i) a clearer sketch of the proof ii) explicit use of sub-Gaussian tail bounds iii) discussion of the candidate set size and its role
>
> > The theoretical results assume linear-Gaussian SEMs (Assumption A5),...
>
> We respectfully disagree with the concern that Assumption A5 (linear-Gaussian SEM) is introduced too late. The assumption is already stated at the very beginning of the theoretical section (Sec. 4, “Preliminaries”), where we define the likelihood model and the BIC score. This is the natural and standard location for introducing distributional assumptions, since all the theoretical guarantees rely on the empirical score $S_n(A)$ defined immediately afterwards. We agree that this assumption limits identifiability to the Markov Equivalence Class level which we explicitly state at the end of the same paragraph.
>
> > The authors should clarify in the introduction that continuous optimization methods ...
>
> We thank the reviewer for the suggestion. Continuous optimization methods such as NOTEARS, GOLEM, and GraN-DAG indeed rely on score functions derived from additive noise or functional causal models. Their objective functions combine a likelihood term with a sparsity or acyclicity penalty, making them conceptually aligned with functional causal model formulations rather than purely score-based search like GES. Our paper already mentions these methods in the related work section.
>
> > Section 3 defers all methodology details to the appendix. While the high-level summary is adequate...
>
> We appreciate the reviewer’s point. Our intention in Section~3 was to keep the main paper concise by presenting the high-level pipeline while placing the step-by-step procedural details in the appendix, consistent with space constraints. The core components needed to understand and reproduce the method state representation, action set, reward structure, warm-starting, and the champion-challenger mechanism are already fully described in the main text. The appendix expands these into implementation-level specifics.
>
> > Distinction from standard warm-start methods: How does warm-start + RL agent differ from simply warm-starting GES...
>
> Please refer  to Appendix A.6 - ``Warm-started GES as a deterministic greedy policy."
>
> >  Relationship to RL-BIC2: How does this work differ from RL-BIC2? Both use RL for causal discovery, ...
>
> RL-BIC2 and our method differ in both design and theory. 1) RL-BIC2 has no candidate-set selection or finite-sample guarantees
> Our Theorem 3 requires: i) Fixed finite candidate set ii) Lipschitz and Gaussian assumptions iii)Statistical concentration of BIC differences. This structure does not exist in RL-BIC2, making Theorem 3 non-transferable. 2) RL-BIC2 uses episodic scoring; we use stepwise reward shaping

---

> ### Author Response · Authors · 2025-12-03
> **Rebuttal (part 3/3)**
>
> Benchmarks in the literature report DAG-level metrics: The community standard (GES, NOTEARS, GOLEM, GraN-DAG, CORL, RL-BIC2) evaluates:
> i) SHD ii) TPR/FDR even for Gaussian or ANM models. We have followed this convention for comparability. On MEC identifiability and the role of interventions,  We fully agree with the reviewer that, under linear-Gaussian SEMs, the DAG is identifiable only up to its Markov equivalence class (MEC). Formally, let
> $$
> \mathrm{MEC}(G) := \{G' \in \mathcal{G} : G' \text{ has the same skeleton and v-structures as } G \}.
> $$
> It is well known that all DAGs in $\mathrm{MEC}(G)$ induce the same observational distribution (Spirtes et al., 2000), and any score equivalent criterion such as BIC satisfies
> $$
> J(G) = J(G') \qquad \forall\, G' \in \mathrm{MEC}(G).
> $$
> Hence, observational BIC selection can at best identify $\mathrm{MEC}(G^\star)$, not $G^\star$ itself.
>
> Interventional data fundamentally changes this picture. If we denote by $\mathrm{MEC}^{\mathcal{I}}(G)$ the interventional equivalence class under a set $\mathcal{I}$ of interventions, then standard results (Hauser and Bühlmann, 2012) imply
> $$
> |\mathrm{MEC}^{\mathcal{I}}(G)| \le |\mathrm{MEC}(G)|,
> $$
> with strict inequality whenever an intervention removes an orientation ambiguity. That is, as more diverse interventions are applied, the equivalence class shrinks, eventually to a singleton under sufficiently rich $\mathcal{I}$. In the limit, this yields point-identifiability of the true DAG.
>
> However, in our setting we operate purely in the observational regime and do not perform any interventions. Therefore the theoretical guarantees in Sections 4.1- 4.3 are necessarily restricted to identifiability at the MEC level rather than the exact DAG. This is also why our analysis focuses on score-based selection rather than exact edge-orientation recovery. Nevertheless, we report DAG-level metrics (SHD, TPR, FDR) to remain consistent with the literature (GES, NOTEARS, GOLEM, GraN-DAG, RL-BIC2),

---

### Official Review · Reviewer_cnVw · 2025-11-01

**Soundness:** 3
**Presentation:** 1
**Contribution:** 2
**Rating:** 2
**Confidence:** 5

**Summary:**

The authors propose a DDQN-based causal discovery method. Their method learns to generate causal graphs that improve the BIC score. Furthermore, they incorporate design choices in their algorithm that provide some theoretical assurances: 1) the output of the agent will have a better or equal score than the original one, 2) an initial graph estimate improves search for good solutions, and 3) good candidates are selected with high probability. They verify the 3rd theorem empirically and put the overall empirical performance of their algorithm in context with existing baselines.Soundness:

**Strengths:**

-Tackling theoretical questions in Reinforcement Learning (RL) is a significant and sparse area of research, and the paper's engagement with this aspect is a clear strength. This focus on theoretical underpinnings helps to advance the field by providing a more robust foundation for future work in RL for causal discovery.

-It's a promising idea to enhance RL for causal discovery by initiating the process with a graph determined by another algorithm. As the authors highlight in their theoretical discussion, this approach could significantly reduce the number of learning episodes required, making the process more efficient.

-The paper's inclusion of an extensive baseline comparison is a notable strength, particularly with the introduction of a comprehensive score.  The use of a comprehensive score further enhances the clarity and interpretability of these comparisons.

**Weaknesses:**

-While the algorithm effectively combines existing elements, its originality is somewhat constrained by the straightforward application of established concepts.

-While the paper touches upon foundational and some contemporary literature, a more comprehensive exploration of recent advancements—such as GFlow Nets [1], RL for causal discovery [2], and other state-of-the-art methods [3]—would greatly enhance the understanding of its impact and relevance within the field.

-Given the inherent design of the algorithm, Theorem 1 seems to be a rather straightforward outcome.

-The current bound in Theorem 2 might be perceived as less impactful due to its considerable looseness (e.g., Amax=10, d=10, 1/π = 1e^23, even for a small graph). It would be great to rephrase the theorem to underscore the critical role of a better initial graph in significantly tightening this value. For improved understanding, it would also be beneficial to include Lemma 1 within the main text.
The soundness of Theorem 3 is challenging to verify. While other theorems also lack some detail, the extent of this issue here makes both the theorem and its proof incredibly difficult to follow. For example, the loose introduction of A^diamond_n significantly obscures the proof.

-Refining the presentation of the mathematical concepts, background information, and general structure is recommended. For example, a more thorough introduction of symbols within the main text, rather than solely relying on an appendix table, would greatly enhance clarity.

-The assumptions seem to largely mirror the algorithm's design, which might prompt questions about their broader purpose and the generalizability of the theoretical findings beyond this particular algorithm. Clarification here would be beneficial.

-While it is beneficial to include numerous benchmarks, the claim of state-of-the-art (SOTA) performance appears to be based on inconclusive results, as several prominent SOTA methods (e.g., DCDI and ENCO) are not included in the comparison.

-In the results section, several statements appear to lack direct support from the presented data. For instance, the method does not seem to outperform GES on the Asia dataset as stated, and on smaller datasets, its performance often mirrors that of GES. Additionally, in some Gran-DAG configurations, the proposed method occasionally leads to a decrease in results.

[1] Deleu, T., Góis, A., Emezue, C., Rankawat, M., Lacoste-Julien, S., Bauer, S. &amp; Bengio, Y.. (2022). Bayesian structure learning with generative flow networks. Proceedings of the Thirty-Eighth Conference on Uncertainty in Artificial Intelligence, in Proceedings of Machine Learning Research.
[2] Sauter, A., Botteghi, N., Acar, E., & Plaat, A. (2024, May). CORE: Towards Scalable and Efficient Causal Discovery with Reinforcement Learning. In Proceedings of the 23rd International Conference on Autonomous Agents and Multiagent Systems
[3] Lippe, P., Cohen, T., & Gavves, E. Efficient Neural Causal Discovery without Acyclicity Constraints. In International Conference on Learning Representations.

**Questions:**

Q: I would be curious about your insights into why sometimes adding this algorithm on top of Gran-DAG leads to worse results. Do you think this could mean that either BIC or the composite score you define are not suitable for evaluation downstream performance?

---

> ### Author Response · Authors · 2025-12-03
> **Rebuttal (part 1/3)**
>
> We thank the reviewer for the feedback!
>
> > Originality is constrained by straightforward application of established concepts
>
> Our approach does not claim a fully two-sided learning game where both agents update their strategies. Instead, the warm-start graph (GES/GraN-DAG) plays the role of a fixed incumbent, and the DDQN agent plays the role of a challenger attempting to improve upon it.
>
> > While the paper touches upon foundational and some contemporary literature, a more comprehensive exploration of recent advancements—such as GFlow Nets...
>
> We thank the reviewer for these helpful suggestions. We agree that positioning our method relative to recent developments will improve the manuscript, and we have incorporated a short dedicated paragraph in the Related Work section summarising the distinctions.
>
> GFlowNets (Deleu et al., 2022) perform Bayesian structure learning by sampling DAGs from an approximate posterior via a flow-matching objective. Although both approaches explore graph space, the goals and mechanisms differ: GFlowNets aim to approximate a full posterior, whereas DDQN-CD optimises a likelihood-based score (BIC) through a champion–challenger Q-learning agent and provides finite-sample guarantees on selecting from a candidate set. Our framework could in fact warm-start from a GFlowNet MAP posterior sample, making the methods complementary rather than competing.
>
> CORE (Sauter et al., 2024) is a scalable RL approach for causal discovery, but its emphasis is on efficient exploration heuristics rather than theoretical guarantees. In contrast, our contribution is to show that, given any warm-start, DDQN-CD provides non-degradation and finite-sample selection guarantees.
>
> > Given the inherent design of the algorithm, Theorem 1 seems to be a rather straightforward outcome.
>
> We agree that Theorem 1 is conceptually simple: it formalises the fact that the
> algorithm returns
> $$
> G_{\mathrm{out}}=\arg\max \{ S_n(\hat{G}),\, S_n(\tilde{G}) \},
> $$
> and therefore cannot worsen the score relative to the warm-start graph.
> Our intent in including this theorem is not to claim mathematical complexity,
> but rather to \emph{make explicit} the safety guarantee that many RL-based
> causal discovery methods implicitly rely on but do not state: the agent can only refine an incumbent solution and never degrade it under the chosen scoring criterion. Theorem 1 also serves as the foundational building block for the subsequent
> results: both Theorem 2 (warm-start efficiency) and Theorem 3 (finite-sample selection consistency) rely on an explicit statement that the algorithm obeys a champion-challenger structure. We have added a note in the revised pdf.
>
> > The current bound in Theorem 2 might be perceived as less impactful due to its considerable looseness......
>
> We agree that the bound in Theorem 2 is conservative; this is a consequence of providing a fully general, worst-case guarantee over all improving trajectories. As in standard RL sample-complexity analysis, we bound the probability of following the \emph{shortest strictly improving path}, since including all suboptimal paths requires modeling score fluctuations and Q-value deviations for every non-monotone move an analysis that becomes exponentially complex and obscures the main insight of the result. Our goal was to provide a guaranteed \emph{lower bound} on hitting time rather than a tight estimate.
> Note that Theorem 2 should be viewed as a principled baseline guarantee rather than a precise characterization of empirical behavior, which in practice is much tighter due to the smoother BIC landscape around good warm starts.
>
> > Refining the presentation of the mathematical concepts, background information, and general structure...
>
> In the revised version we have  added a new section namely ``Background and Preliminaries" which includes  (i) a short subsection on DAGs, CPDAGs, Markov equivalence and the causal discovery problem; (ii) an RL preliminaries subsection (MDP, state, action, reward, value function, $\varepsilon$-greedy policy); (iii) a problem-setup paragraph explicitly stating that states are DAGs, actions are edge add/remove/reverse operations, and rewards are BIC score differences.

---

> ### Author Response · Authors · 2025-12-03
> **Rebuttal (part 2/3)**
>
> > The assumptions seem to largely mirror the algorithm's design, which might prompt questions...
>
> Our intent is to abstract the minimal structural properties needed to prove the guarantees, not to tailor assumptions just to our methodology.
> A1–A4 (RL + champion–challenger).
> These assumptions are generic to any finite-horizon RL refinement scheme over DAGs:
> A1 (finite feasibility) only requires acyclicity and a finite edge budget. Any method that edits graphs under a sparsity constraint would satisfy this.
> A2 (persistent exploration) is a standard RL assumption (e.g., $\epsilon$-greedy or softmax policies).
> A3 (warm start) does not require GES or GraN-DAG specifically; any initial DAG (including domain-expert priors or other neural methods) suffices.
> A4 (champion–challenger selection) can be used by any method that maintains a running incumbent score.
> A5–A6 (Gaussian + Lipschitz).
> These are primarily needed for Theorem 3. They are standard in the BIC literature and could be relaxed to more general sub-Gaussian designs or other Lipschitz-type conditions. The assumptions are not specific to Double DQN; similar guarantees could apply to other RL schemes that satisfy A1–A4. Our framework is modular in the opponent choice and may generalize to different scoring rules if they satisfy analogues of A5–A6.
>
> > While it is beneficial to include numerous benchmarks, the claim of state-of-the-art (SOTA) performance appears to be based on...
>
> We thank the reviewer for raising this point. Regarding DCDI, we would like to clarify that DCDI is explicitly designed for the interventional data setting: its likelihood factorization, soft-intervention modeling, and training objective fundamentally rely on intervention-specific terms. Since our work focuses strictly on observational data, applying DCDI would violate its modeling assumptions and in our preliminary attempts yielded degenerate or non-identifiable solutions. For this reason, it is not an appropriate baseline for the setting we study.
>
> For ENCO, although often categorized as a gradient-based causal discovery method, its own limitations section states that its theoretical guarantees require interventions on all variables and that performance significantly depends on the presence and quality of intervention data. ENCO additionally models edge orientations as continuous parameters unconstrained by acyclicity, which is beneficial in the mixed observational-interventional regime but can behave unreliably in the purely observational linear-Gaussian setting considered in our work. As such, ENCO is not directly comparable to our method’s domain and including it would not provide a fair or informative comparison. DCDI and ENCO are not well-aligned with the observational-only scope of this paper.
>
> > In the results section, several statements appear to lack direct support from the presented data....
>
> We examined the reviewer’s concern and would like to clarify that a decrease relative to GraN-DAG occurs only on a single dataset Lucas. On every other dataset, DDQN-CD either matches or improves upon GraN-DAG. Thus, the statement that the method decreases performance ``in some configurations’’ overstates the effect; it is isolated to one dataset rather than a recurring pattern.
>
> For Lucas specifically, GraN-DAG itself performs inconsistently (e.g., high FDR and relatively poor SHD), and the warm-start graph it provides is already suboptimal. Since DDQN-CD is a refinement method that builds on the given initial structure, any weaknesses in the warm-start propagate into the refinement stage. The slight drop in the Composite Score is therefore a direct consequence of the unstable GraN-DAG initialization on this particular dataset, not a failure of the RL-based refinement mechanism.

---

> ### Author Response · Authors · 2025-12-03
> **Rebuttal (part 3/3)**
>
> >  I would be curious about your insights into why sometimes adding this algorithm on top of Gran-DAG leads to worse results...
>
> Our theoretical guarantees are explicitly derived with respect to the BIC score, since BIC is the canonical likelihood-based objective widely used in causal discovery (e.g., GES, RL-BIC, CORL). However, for empirical evaluation, it is standard practice to assess performance using structural metrics such as SHD, FDR, and TPR. These metrics capture complementary aspects of correctness: SHD measures adjacency-level accuracy, FDR penalizes false positives, and TPR rewards correct edge recovery. In practice, different algorithms tend to excel on different metrics, making it difficult to obtain a single, fair indicator of overall performance. To address this, we introduce a Composite Score that aggregates SHD, FDR, and TPR into a single evaluation signal. This score is explicitly designed to reflect multi-objective structural quality across sparsity, precision, and recall. Importantly, although our theoretical bound is formulated only w.r.t. BIC, our experiments consistently show that the proposed framework either matches or improves the Composite Score. This empirically confirms that optimizing BIC, when combined with our RL-based exploration and pruning translates into better overall structural behavior, even when individual metrics (e.g., SHD) may fluctuate due to finite-sample trade-offs.

---

### Author Response · Authors · 2025-12-03
**Dear Reviewers (rebuttal)**

# **General Response to All Reviewers**

We sincerely thank all reviewers for their thoughtful and constructive feedback. Their comments significantly helped us improve the clarity, completeness, and technical depth of the paper. We have revised the manuscript as per the requirement, and **all changes in the updated PDF are highlighted in *blue*** for easy inspection.

Across the reviews, we identified several recurring themes - clarity of methodology, completeness of baseline comparisons, placement of assumptions, theoretical detail, and empirical interpretation. We address each systematically:

* **Clarity & Methodological Detail:**
  We expanded Section 3 with clearer algorithmic descriptions, added missing reinforcement-learning preliminaries, explained reward normalization, and strengthened the discussion on how our RL refinement differs from GES-like greedy search.

* **Assumptions & Identifiability:**
  We now explicitly state the linear-Gaussian assumption (A5) *upfront* in the theory section, clarify that all guarantees are Markov Equivalence Class-level (**MEC-level**), and add a dedicated paragraph connecting MEC, observational identifiability, and absence of interventions.

* **Expanded Related Work:**
  We incorporated reviewers’ suggestions by adding comparisons with **GFlowNets**, **CORE**,  and **neural causal models**, clarifying conceptual differences and complementarities.

* **Additional Baseline Discussion:**
  We now justify the baseline choices more clearly (including why DCDI/ENCO are not directly applicable to purely observational settings), and annotate table captions explaining why some baselines do not appear for large-scale datasets.

* **Theoretical Clarifications:**
  We tightened the proof sketches of Theorems 2 and 3, added sub-Gaussian arguments explicitly, clarified constants, and explained why shortest improving paths are used in hitting-time bounds.

* **Empirical Interpretation:**
  We  clarified cases where improvement is metric-dependent, added comments on SHD limitations, and provided dataset-specific explanations (e.g., Lucas) to avoid misinterpretation.

---

### Meta-Review · Area_Chair_W9QH · 2026-01-06

**Summary:**

The submission proposes DDQN-CD, a DDQN-based reinforcement-learning framework that refines a warm-start causal graph (e.g., GES or GraN-DAG) via edge edit actions with BIC-shaped rewards, and claims (i) non-degradation relative to the warm start, (ii) faster convergence with better initialization, and (iii) finite-sample high-probability selection among collected candidates.

**Reviewer Concerns:**

Reviewers appreciate the motivation of adding theory to RL-based causal discovery and the broad benchmark coverage, and the rebuttal credibly improves presentation (added preliminaries, clearer algorithm description), clarifies linear-Gaussian/MEC-level scope, and strengthens related-work positioning and baseline justifications. However, the core concerns remain: the novelty and necessity of RL (and DDQN specifically) over simpler warm-started local search is still not convincingly demonstrated; key theoretical results are either largely by construction (Theorem 1) or too loose to verify from the main paper despite revisions (Theorems 2–3); and empirical gains appear marginal or metric-dependent on multiple datasets, with missing/limited large-scale comparisons and lingering misalignment between MEC identifiability and DAG-level evaluation.

**Reviewer Scores:**

Reviewer cnVw: likely stays at 2.
They were very confident (confidence 5) and their biggest issues weren’t just missing explanations—they questioned originality, found Theorem 2 too loose, found Theorem 3 hard to verify, and believed the empirical claims were sometimes not supported. The rebuttal helps on presentation/related work/baseline justification and clarifies the “Lucas-only” degradation point, but it doesn’t really fix the perceived looseness of Theorem 2 and Theorem 1, and the novelty concern remains.

Reviewer ZRcW: likely 2->4.
The rebuttal addresses most concerns, but the empirical “practical significance” and baseline coverage concerns still remain.

Reviewer 2q24: likely stays at 2.
Critical concerns on contribution/empirical value may remain unsolved.

---

### Decision · Program_Chairs · 2026-01-26

Reject